# A Theoretical Study on Solving Continual Learning

**Gyuhak Kim**[*1], **Changnan Xiao**[*2], **Tatsuya Konishi**[† 3], **Zixuan Ke**[1], **Bing Liu**[‡ 1]

[1] University of Illinois at Chicago
[2] ByteDance
[3] KDDI Research

## Abstract

Continual learning (CL) learns a sequence of tasks incrementally. There are two popular CL settings, *class incremental learning* (CIL) and *task incremental learning* (TIL). A major challenge of CL is *catastrophic forgetting* (CF). While several techniques are available to effectively overcome CF for TIL, CIL remains to be challenging due to the additional difficulty of *inter-task class separation*. So far little theoretical work has been done to provide a *principled guidance* and *necessary and sufficient* conditions for solving the CIL problem. This paper performs such a study. It first probabilistically decomposes the CIL problem into two sub-problems: *within-task prediction* (WP) and *task-id prediction* (TP). It further proves that TP is correlated with *out-of-distribution* (OOD) detection. The key *result* is that regardless of whether WP and TP or OOD detection are defined explicitly or implicitly by a CIL algorithm, good WP and good TP or OOD detection are *necessary* and *sufficient* for good CIL performances. Additionally, TIL is simply WP. Based on the theoretical result, new CIL methods are also designed, which outperform strong baselines in both CIL and TIL settings by a large margin.[4]

## 1 Introduction

Continual learning aims to incrementally learn a sequence of tasks [1]. Each task consists of a set of classes to be learned together. A major challenge of CL is *catastrophic forgetting* (CF). Although a large number of CL techniques have been proposed, they are mainly empirical. Limited theoretical research has done on how to solve CL. This paper performs such a theoretical study about the necessary and sufficient conditions for effective CL. There are two main CL settings that have been extensively studied: *class incremental learning* (CIL) and *task incremental learning* (TIL) [2]. In CIL, the learning process builds a single classifier for all tasks/classes learned so far. In testing, a test instance from any class may be presented for the model to classify. No prior task information (e.g., task-id) of the test instance is provided. Formally, CIL is defined as follows.

**Class incremental learning** (CIL). CIL learns a sequence of tasks, $1, 2, ..., T$. Each task $k$ has a training dataset $\mathcal{D}_k = \{(x_k^i, y_k^i)_{i=1}^{n_k}\}$, where $n_k$ is the number of data samples in task $k$, and $x_k^i \in \mathbf{X}$ is an input sample and $y_k^i \in \mathbf{Y}_k$ (the set of all classes of task $k$) is its class label. All $\mathbf{Y}_k$'s are disjoint $(\mathbf{Y}_k \cap \mathbf{Y}_{k'} = \emptyset, \forall k \neq k')$ and $\bigcup_{k=1}^{T} \mathbf{Y}_k = \mathbf{Y}$. The goal of CIL is to construct a single predictive function or classifier $f : \mathbf{X} \rightarrow \mathbf{Y}$ that can identify the class label $y$ of each given test instance $x$.

In the TIL setup, each task is a separate classification problem. For example, one task could be to classify different breeds of dogs and another task could be to classify different types of animals (the

---

[*]Equal contribution

[†]The work was done when this author was visiting Bing Liu's group at University of Illinois at Chicago

[‡]Correspondance author. Bing Liu <liub@uic.edu>

[4]The code is available at https://github.com/k-gyuhak/WPTP

tasks may not be disjoint). One model is built for each task in a shared network. In testing, the task-id of each test instance is provided and the system uses only the specific model for the task (dog or animal classification) to classify the test instance. Formally, TIL is defined as follows.

**Task incremental learning** (TIL). TIL learns a sequence of tasks, $1, 2, ..., T$. Each task $k$ has a training dataset $\mathcal{D}_k = \{((x_k^i, k), y_k^i)_{i=1}^{n_k}\}$, where $n_k$ is the number of data samples in task $k \in \mathbf{T} = \{1, 2, ..., T\}$, and $x_k^i \in \mathbf{X}$ is an input sample and $y_k^i \in \mathbf{Y}_k \subset \mathbf{Y}$ is its class label. The goal of TIL is to construct a predictor $f : \mathbf{X} \times \mathbf{T} \to \mathbf{Y}$ to identify the class label $y \in \mathbf{Y}_k$ for $(x, k)$ (the given test instance $x$ from task $k$).

Several techniques are available to effectively overcome CF for TIL (with almost no CF) [3, 4]. However, CIL remains to be highly challenging due to the additional problem of **Inter-task Class Separation** (ICS) (establishing decision boundaries between the classes of the new task and the classes of the previous tasks) because the previous task data are not accessible. To our knowledge, the ICS problem has not been discussed before but it is critical for the success of CIL. Before discussing the proposed work, we recall the *closed-world* assumption made by traditional machine learning, i.e., *the classes seen in testing must have been seen in training* [1, 5]. However, in many applications, there are unknowns in testing, which is called the *open world* setting [1, 5]. In open world learning, the training (or known) classes are called *in-distribution* (IND) classes. A classifier built for the open world can (1) classify test instances of training/IND classes to their respective classes, which is called *IND prediction*, and (2) detect test instances that do not belong to any of the IND/known classes but some unknown or *out-of-distribution* (OOD) classes, which is called *OOD detection*. Many OOD detection algorithms can perform both IND prediction and OOD detection [6, 7, 8, 9]. The commonality of OOD detection and CIL is that they both need to consider future unknowns.

This paper conducts a theoretical study of CIL, which is applicable to any CIL classification model. Instead of focusing on the traditional PAC generalization bound [10, 11], we focus on how to solve the CIL problem. We first decompose the CIL problem into two sub-problems in a probabilistic framework: *Within-task Prediction* (WP) and *Task-id Prediction* (TP). WP means that the prediction for a test instance is only done within the classes of the task to which the test instance belongs, which is basically the TIL problem. TP predicts the task-id. TP is needed because in CIL, task-id is not provided in testing. This paper then proves based on the popular cross-entropy loss that (1) the CIL performance is bounded by WP and TP performances, and (2) TP and task OOD detection performance bound each other (which connects CL and OOD detection). The key result is that regardless of whether WP and TP or OOD detection are defined explicitly or implicitly by a CIL algorithm, good WP and good TP or OOD detection are *necessary* and *sufficient* conditions for good CIL performances. This result is applicable to both batch/offline and online CIL and to CIL problems with blurry task boundaries. The intuition is also quite simple because if a CIL model is perfect at detecting OOD samples for each task (which solves the ICS problem), then CIL is reduced to WP.

This theoretical result provides a principled guidance for solving the CIL problem, i.e., to help design better CIL algorithms that can achieve strong WP and TP performances. Since WP is basically IND prediction for each task and most OOD techniques perform both IND prediction and OOD detection, to achieve good CIL accuracy, a strong OOD performance for each task is necessary.

Based on the theoretical guidance, several new CIL methods are designed, including techniques based on the integration of a TIL method and an OOD detection method for CIL, which outperform strong baselines in both the CIL and TIL settings by a large margin. This combination is particularly attractive because TIL has achieved no forgetting, and we only need a strong OOD technique that can perform both IND prediction and OOD detection to learn each task to achieve strong CIL results.

## 2 Related Work

Although numerous CL techniques have been proposed, little theoretical work has been done on how to solve CL. Existing methods are mainly empirical and belong to several categories. Using regularization [12, 13] to minimize changes to model parameters learned from previous tasks is a popular approach [14, 15, 16, 17, 18, 19, 20, 21, 22, 23, 24, 25, 26]. Memorizing some old examples and using them to jointly train the new task is another popular approach (called *replay*) [27, 28, 29, 30, 31, 32, 33, 34, 35, 36, 37, 38, 39, 40]. Some systems learn to generate pseudo training data of old tasks to be used in replay, called *pseudo-replay* [41, 42, 43, 44, 45, 46, 21, 47, 48]. Orthogonal projection learns each task in an orthogonal space to other tasks [49, 50, 51].

*Parameter isolation* is yet another popular approach, which makes different subsets (which may overlap) of the network parameters dedicated to different tasks using masks [3, 52, 53, 4, 54]. This approach is particularly suited for TIL. Several methods have almost completely overcome forgetting. HAT [3] and CAT [52] protect previous tasks by masking the important parameters to those tasks. PackNet [53], CPG [54] and SupSup [4] find an isolated sub-network for each task. HyperNet [55] initializes task-specific parameters conditioned on task-id. ADP [56] decomposes parameters into shared and adaptive parts to construct an order robust TIL system. CCLL [57] uses task-adaptive calibration in convolution layers. Our methods designed based on the proposed theory make use of two parameter isolation-based TIL methods and two OOD detection methods. A strong OOD detection method CSI in [6] helps produce very strong CIL results. CSI is based on data augmentation [58] and contrastive learning [59]. Excellent surveys of OOD detection include [60, 61].

Some methods have used a TIL method for CIL with an additional task-id prediction technique. iTAML [62] requires each test batch to be from a single task. This is not practical as test samples usually come one by one. CCG [63] builds a separate network to predict the task-id. Expert Gate [64] constructs a separate autoencoder for each task. HyperNet [55] and PR-Ent [65] use entropy to predict the task id. Since none of these papers is a theoretical study, they did not know that strong OOD detection is the key. Our methods based on OOD detection perform dramatically better.

Several theoretical studies have been made on lifelong/continual learning. However, they focus on traditional generalization bound. [10] proposes a PAC-Bayesian framework to provide a learning bound on expected error in future tasks by the average loss on the observed tasks. The work in [66] studies the generalization error by task similarity and [11] studies the dependence of generalization error on sample size or number of tasks including forward and backward transfer. [67] shows that orthogonal gradient descent gives a tighter generalization bound than SGD. Our work is very different as we focus on how to solve the CIL problem, which is orthogonal to the existing theoretical analysis.

## 3   CIL by Within-Task Prediction and Task-ID Prediction

This section presents our theory. It first shows that the CIL performance improves if the within-task prediction (WP) performance and/or the task-id prediction (TP) performance improve, and then shows that TP and OOD detection bound each other, which indicates that CIL performance is controlled by WP and OOD detection. This connects CL and OOD detection. Finally, we study the necessary conditions for a good CIL model, which includes a good WP, and a good TP (or OOD detection).

### 3.1   CIL Problem Decomposition

This sub-section first presents the assumptions made by CIL based on its definition and then proposes a decomposition of the CIL problem into two sub-problems. A CL system learns a sequence of tasks $\{(\mathbf{X}_k, \mathbf{Y}_k)\}_{k=1,\dots,T}$, where $\mathbf{X}_k$ is the domain of task $k$ and $\mathbf{Y}_k$ are classes of task $k$ as $\mathbf{Y}_k = \{\mathbf{Y}_{k,j}\}$, where $j$ indicates the $j$th class in task $k$. Let $\mathbf{X}_{k,j}$ to be the domain of $j$th class of task $k$, where $\mathbf{X}_k = \bigcup_j \mathbf{X}_{k,j}$. For accuracy, we will use $x \in \mathbf{X}_{k,j}$ instead of $\mathbf{Y}_{k,j}$ in probabilistic analysis. Based on the definition of class incremental learning (CIL) (Sec. 1), the following assumptions are implied,

**Assumption 1.** The domains of classes of the same task are disjoint, i.e., $\mathbf{X}_{k,j} \cap \mathbf{X}_{k,j'} = \emptyset, \forall j \neq j'$.

**Assumption 2.** The domains of tasks are disjoint, i.e., $\mathbf{X}_k \cap \mathbf{X}_{k'} = \emptyset, \forall k \neq k'$.

For any ground event $D$, the goal of a CIL problem is to learn $\mathbf{P}(x \in \mathbf{X}_{k,j}|D)$. This can be decomposed into two probabilities, *within-task IND prediction* (WP) probability and *task-id prediction* (TP) probability. WP probability is $\mathbf{P}(x \in \mathbf{X}_{k,j}|x \in \mathbf{X}_k, D)$ and TP probability is $\mathbf{P}(x \in \mathbf{X}_k|D)$. We can rewrite the CIL problem using WP and TP based on the two assumptions,

$$\mathbf{P}(x \in \mathbf{X}_{k_0,j_0}|D) = \sum_{k=1,\dots,n} \mathbf{P}(x \in \mathbf{X}_{k,j_0}|x \in \mathbf{X}_k, D)\mathbf{P}(x \in \mathbf{X}_k|D) \tag{1}$$

$$= \mathbf{P}(x \in \mathbf{X}_{k_0,j_0}|x \in \mathbf{X}_{k_0}, D)\mathbf{P}(x \in \mathbf{X}_{k_0}|D) \tag{2}$$

where $k_0$ means a particular task and $j_0$ a particular class in the task.

Some remarks are in order about Eq. 2 and our subsequent analysis to set the stage.

*Remark* 1. Eq. 2 shows that if we can improve either the WP or TP performance, or both, we can improve the CIL performance.

*Remark* 2. It is important to note that our theory is not concerned with the learning algorithm or the training process, but we will propose some concrete learning algorithms based on the theoretical result in the experiment section.

*Remark* 3. Note that the CIL definition and the subsequent analysis are applicable to tasks with any number of classes (including only one class per task) and to online CIL where the training data for each task or class comes gradually in a data stream and may also cross task boundaries (blurry tasks [68]) because our analysis is based on an already-built CIL model after training. Regarding blurry task boundaries, suppose dataset 1 has classes {dog, cat, tiger} and dataset 2 has classes {dog, computer, car}. We can define task 1 as {dog, cat, tiger} and task 2 as {computer, car}. The shared class *dog* in dataset 2 is just additional training data of *dog* appeared after task 1.

*Remark* 4. Furthermore, CIL = WP * TP in Eq. 2 means that when we have WP and TP (defined either explicitly or implicitly by implementation), we can find a corresponding CIL model defined by WP * TP. Similarly, when we have a CIL model, we can find the corresponding underlying WP and TP defined by their probabilistic definitions.

In the following sub-sections, we develop this further concretely to derive the sufficient and necessary conditions for solving the CIL problem in the context of cross-entropy loss as it is used in almost all supervised CL systems.

## 3.2 CIL Improves as WP and/or TP Improve

As stated in Remark 2 above, the study here is based on a *trained CIL model* and not concerned with the algorithm used in training the model. We use cross-entropy as the performance measure of a trained model as it is the most popular loss function used in supervised CL. For experimental evaluation, we use *accuracy* following CL papers. Denote the cross-entropy of two probability distributions $p$ and $q$ as

$$H(p, q) \overset{def}{=} -\mathbb{E}_p[\log q] = -\sum_i p_i \log q_i. \tag{3}$$

For any $x \in \mathbf{X}$, let $y$ to be the CIL ground truth label of $x$, where $y_{k_0,j_0} = 1$ if $x \in \mathbf{X}_{k_0,j_0}$ otherwise $y_{k,j} = 0, \forall (k, j) \neq (k_0, j_0)$. Let $\tilde{y}$ be the WP ground truth label of $x$, where $\tilde{y}_{k_0,j_0} = 1$ if $x \in \mathbf{X}_{k_0,j_0}$ otherwise $\tilde{y}_{k_0,j} = 0, \forall j \neq j_0$. Let $\bar{y}$ be the TP ground truth label of $x$, where $\bar{y}_{k_0} = 1$ if $x \in \mathbf{X}_{k_0}$ otherwise $\bar{y}_k = 0, \forall k \neq k_0$. Denote

$$H_{WP}(x) = H(\tilde{y}, \{\mathbf{P}(x \in \mathbf{X}_{k_0,j} | x \in \mathbf{X}_{k_0}, D)\}_j), \tag{4}$$

$$H_{CIL}(x) = H(y, \{\mathbf{P}(x \in \mathbf{X}_{k,j} | D)\}_{k,j}), \tag{5}$$

$$H_{TP}(x) = H(\bar{y}, \{\mathbf{P}(x \in \mathbf{X}_k | D)\}_k) \tag{6}$$

where $H_{WP}$, $H_{CIL}$ and $H_{TP}$ are the cross-entropy values of WP, CIL and TP, respectively. We now present our first theorem. The theorem connects CIL to WP and TP, and suggests that by having a good WP or TP, the CIL performance improves as the upper bound for the CIL loss decreases.

**Theorem 1.** *If $H_{TP}(x) \leq \delta$ and $H_{WP}(x) \leq \epsilon$, we have $H_{CIL}(x) \leq \epsilon + \delta$.*

The detailed proof is given in Appendix A.1. This theorem holds regardless of whether WP and TP are trained together or separately. When they are trained separately, if WP is fixed and we let $\epsilon = H_{WP}(x)$, $H_{CIL}(x) \leq H_{WP}(x) + \delta$, which means if TP is better, CIL is better. Similarly, if TP is fixed, we have $H_{CIL}(x) \leq \epsilon + H_{TP}(x)$. When they are trained concurrently, there exists a functional relationship between $\epsilon$ and $\delta$ depending on implementation. But no matter what it is, when $\epsilon + \delta$ decreases, CIL gets better.

Theorem 1 holds for any $x \in \mathbf{X} = \bigcup_k \mathbf{X}_k$ that satisfies $H_{TP}(x) \leq \delta$ or $H_{WP}(x) \leq \epsilon$. To measure the overall performance under expectation, we present the following corollary.

**Corollary 1.** *Let $U(\mathbf{X})$ represents the uniform distribution on $\mathbf{X}$. i) If $\mathbb{E}_{x \sim U(\mathbf{X})}[H_{TP}(x)] \leq \delta$, then $\mathbb{E}_{x \sim U(\mathbf{X})}[H_{CIL}(x)] \leq \mathbb{E}_{x \sim U(\mathbf{X})}[H_{WP}(x)] + \delta$. Similarly, ii) $\mathbb{E}_{x \sim U(\mathbf{X})}[H_{WP}(x)] \leq \epsilon$, then $\mathbb{E}_{x \sim U(\mathbf{X})}[H_{CIL}(x)] \leq \epsilon + \mathbb{E}_{x \sim U(\mathbf{X})}[H_{TP}(x)]$.*

The proof is given in Appendix A.2. The corollary is a direct extension of Theorem 1 in expectation. The implication is that given TP performance, CIL is positively related to WP. The better the WP is, the better the CIL is as the upper bound of the CIL loss decreases. Similarly, given WP performance, a better TP performance results in a better CIL performance. Due to the positive relation, we can improve CIL by improving either WP or TP using their respective methods developed in each area.

## 3.3 Task Prediction (TP) to OOD Detection

Building on Eq. 2, we have studied the relationship of CIL, WP and TP in Theorem 1. We now connect TP and OOD detection. They are shown to be dominated by each other to a constant factor.

We again use cross-entropy $H$ to measure the performance of TP and OOD detection of a trained network as in Sec. 3.2 To build the connection between $H_{TP}(x)$ and OOD detection of each task, we first define the notations of OOD detection. We use $\mathbf{P}'_k(x \in \mathbf{X}_k|D)$ to represent the probability distribution predicted by the $k$th task's OOD detector. Notice that the task prediction (TP) probability distribution $\mathbf{P}(x \in \mathbf{X}_k|D)$ is a categorical distribution over $T$ tasks, while the OOD detection probability distribution $\mathbf{P}'_k(x \in \mathbf{X}_k|D)$ is a Bernoulli distribution. For any $x \in \mathbf{X}$, define

$$H_{OOD,k}(x) = \begin{cases} H(1, \mathbf{P}'_k(x \in \mathbf{X}_k|D)) = -\log \mathbf{P}'_k(x \in \mathbf{X}_k|D), \ x \in \mathbf{X}_k, \\ H(0, \mathbf{P}'_k(x \in \mathbf{X}_k|D)) = -\log \mathbf{P}'_k(x \notin \mathbf{X}_k|D), \ x \notin \mathbf{X}_k. \end{cases} \tag{7}$$

In CIL, the OOD detection probability for a task can be defined using the output values corresponding to the classes of the task. Some examples of the function is a sigmoid of maximum logit value or a maximum softmax probability after re-scaling to 0 to 1. It is also possible to define the OOD detector directly as a function of tasks instead of a function of the output values of all classes of tasks, i.e. Mahalanobis distance. The following theorem shows that TP and OOD detection bound each other.

**Theorem 2.** *i) If $H_{TP}(x) \leq \delta$, let $\mathbf{P}'_k(x \in \mathbf{X}_k|D) = \mathbf{P}(x \in \mathbf{X}_k|D)$, then $H_{OOD,k}(x) \leq \delta, \forall k = 1, \ldots, T$. ii) If $H_{OOD,k}(x) \leq \delta_k, k = 1, \ldots, T$, let $\mathbf{P}(x \in \mathbf{X}_k|D) = \frac{\mathbf{P}'_k(x \in \mathbf{X}_k|D)}{\sum_k \mathbf{P}'_k(x \in \mathbf{X}_k|D)}$, then $H_{TP}(x) \leq (\sum_k \mathbf{1}_{x \in \mathbf{X}_k} e^{\delta_k})(\sum_k 1 - e^{-\delta_k})$, where $\mathbf{1}_{x \in \mathbf{X}_k}$ is an indicator function.*

See Appendix A.3 for the proof. As we use cross-entropy, the lower the bound, the better the performance is. The first statement (i) says that the OOD detection performance improves if the TP performance gets better (i.e., lower $\delta$). Similarly, the second statement (ii) says that the TP performance improves if the OOD detection performance on each task improves (i.e., lower $\delta_k$). Besides, since $(\sum_k \mathbf{1}_{x \in \mathbf{X}_k} e^{\delta_k})(\sum_k 1 - e^{-\delta_k})$ converges to 0 as $\delta_k$'s converge to 0 in order of $O(|\sum_k \delta_k|)$, we further know that $H_{TP}$ and $\sum_k H_{OOD,k}$ are equivalent in quantity up to a constant factor.

In Theorem 1, we studied how CIL is related to WP and TP. In Theorem 2, we showed that TP and OOD bound each other. Now we explicitly give the upper bound of CIL in relation to WP and OOD detection of each task. The detailed proof can be found in Appendix A.4.

**Theorem 3.** *If $H_{OOD,k}(x) \leq \delta_k, k = 1, \ldots, T$ and $H_{WP}(x) \leq \epsilon$, we have*

$$H_{CIL}(x) \leq \epsilon + (\sum_k \mathbf{1}_{x \in \mathbf{X}_k} e^{\delta_k})(\sum_k 1 - e^{-\delta_k}),$$

*where $\mathbf{1}_{x \in \mathbf{X}_k}$ is an indicator function.*

## 3.4 Necessary Conditions for Improving CIL

In Theorem 1, we showed that good performances of WP and TP are sufficient to guarantee a good performance of CIL. In Theorem 3, we showed that good performances of WP and OOD are sufficient to guarantee a good performance of CIL. For completeness, we study the necessary conditions of a well-performed CIL in this sub-section.

**Theorem 4.** *If $H_{CIL}(x) \leq \eta$, then there exist i) a WP, s.t. $H_{WP}(x) \leq \eta$, ii) a TP, s.t. $H_{TP}(x) \leq \eta$, and iii) an OOD detector for each task, s.t. $H_{OOD,k} \leq \eta, k = 1, \ldots, T$.*

The detailed proof is given in Appendix A.5. This theorem tells that if a good CIL model is trained, then a good WP, a good TP and a good OOD detector for each task are always implied. More importantly, by transforming Theorem 4 into its contraposition, we have the following statements: If for any WP, $H_{WP}(x) > \eta$, then $H_{CIL}(x) > \eta$. If for any TP, $H_{TP}(x) > \eta$, then $H_{CIL}(x) > \eta$. If for any OOD detector, $H_{OOD,k}(x) > \eta, k = 1, \ldots, T$, then $H_{CIL}(x) > \eta$. Regardless of whether WP and TP (or OOD detection) are defined explicitly or implicitly by a CIL algorithm, the existence of a good WP and the existence of a good TP or OOD detection are necessary conditions for a good CIL performance.

*Remark* 5. It is important to note again that our study in this section is based on a CIL model that has already been built. In other words, our study tells the CIL designers what should be achieved in the final model. Clearly, one would also like to know how to design a strong CIL model based on the theoretical results, which also considers catastrophic forgetting (CF). One effective method is to make use of a strong existing TIL algorithm, which can already achieve no or little forgetting (CF), and combine it with a strong OOD detection algorithm (as mentioned earlier, most OOD detection methods can also perform WP). Thus, any improved method from the OOD detection community can be applied to CIL to produce improved CIL systems (see Sections 4.3 and 4.4).

Recall in Section 2, we reviewed prior works that have tried to use a TIL method for CIL with a task-id prediction method [55, 64, 62, 63, 65]. However, since they did not know that the key to the success of this approach is a strong OOD detection algorithm, they are quite weak (see Section 4).

## 4 New CIL Techniques and Experiments

Based on Theorem 3, we have designed several new CIL methods, each of which integrates an existing CL algorithm and an OOD detection algorithm. The OOD detection algorithm that we use can perform both within-task IND prediction (WP) and OOD detection. Our experiments have two goals: (1) to show that a good OOD detection method can help improve the accuracy of an existing CIL algorithm, and (2) to fully compare two of these methods (see some others in Sec. 4.5) with strong baselines to show that they outperform the existing strong baselines considerably.

### 4.1 Datasets, CL Baselines and OOD Detection Methods

**Datasets and CIL Tasks.** Four popular benchmark image classification datasets are used, from which six CIL problems are created following recent papers [25, 34, 26]. **(1)** *MNIST* consists of handwritten images of 10 digits with 60,000/10,000 training/testing samples. We create a CIL problem (**M-5T**) of 5 tasks with 2 consecutive classes/digits as a task. **(2)** *CIFAR-10* consists of 32x32 color images of 10 classes with 50,000/10,000 training/testing samples. We create a CIL problem (**C10-5T**) of 5 tasks with 2 consecutive classes as a task. **(3)** *CIFAR-100* consists of 60,000 32x32 color images with 50,000/10,000 training/testing samples. We create two CIL problems by splitting 100 classes into 10 tasks (**C100-10T**) and 20 tasks (**C100-20T**), where each task has 10 and 5 classes, respectively. **(4)** *Tiny-ImageNet* has 120,000 64x64 color images of 200 classes with 500/50 images per class for training/testing. We create two CIL problems by splitting 200 classes into 5 tasks (**T-5T**) and 10 tasks (**T-10T**), where each task has 40 and 20 classes, respectively.

**Baseline CL Methods.** We include different families of CL methods: *regularization*, *replay*, *orthogonal projection*, and *parameter isolation*. MUC [25] and PASS [26] are regularization-based methods. For replay methods, we use LwF [13], iCaRL [29], Mnemonics [69], BiC [32], DER++ [34], and $Co^2L$ [37]. For orthogonal projection, we use OWM [49]. Finally, for parameter isolation, we use CCG [63], HyperNet [55], HAT [3], SupSup [4] (Sup), and PR [65].[5] We use the official codes for the baselines except for $Co^2L$, CCG, and PR. For these three systems, we copy the results from their papers as the code for CCG is not released and we are unable to run $Co^2L$ and PR on our machines.

**OOD Detection Methods**. Two OOD detection methods are used. We combine them with the above existing CL algorithms. Both these methods can also perform *within-task IND prediction* (WP).

**(1). ODIN**: Researchers have proposed several methods to improve the OOD detection performance of a trained network by post-processing [7, 70, 71]. ODIN [7] is a representative method. It adds perturbation to input and applies a temperature scaling to the softmax output of a trained network.

**(2). CSI**: It is a recently proposed OOD detection technique [6] that is highly effective. It is based on data and class label augmentation and supervised contrastive learning [72]. Its rotation data

---

[5]iTAML [62] is not included as it requires a batch of test data from the same task to predict the task-id. When each batch has only one test sample, which is our setting, it is very weak. For example, its CIL accuracy is only 33.5% on C100-10T. Expert Gate (EG) [64] is also very weak. Its CIL accuracy is only 43.2 on M-5T. They are much weaker than many baselines. DER [38] is not included as it expands the network after each task, which is somewhat unfair to other systems as all others do not expand the network. DER can generate a large number of parameters after the last task, e.g., 117.6 millions (M) for C100-20T while our proposed methods require 44.6M (HAT+CSI) and 11.6M (Sup+CSI) (refer to Appendix H). The average accuracy of DER over the 6 CL experiments is 61.4 while our methods achieve 67.9 (HAT+CSI+c) and 64.9 (Sup+CSI+c) (refer to Tab. 3).

augmentations create distributional shifted samples to act as negative data for the original samples for contrastive learning. The details of CSI is given in Appendix D.

## 4.2 Training Details and Evaluation Metrics

**Training Details.** For the backbone structure, we follow [4, 26, 34]. AlexNet-like architecture [73] is used for MNIST and ResNet-18 [74] is used for CIFAR-10. For CIFAR-100 and Tiny-ImageNet, ResNet-18 is also used as CIFAR-10, but the number of channels are doubled to fit more classes. All the methods use the same backbone architecture except for OWM and HyperNet, for which we use their original architectures. OWM uses AlexNet. It is not obvious how to apply the technique to the ResNet structure. HyperNet uses a fully-connected network and ResNet-32 for MNIST and other datasets, respectively. We are unable to change the structure due to model initialization arguments unexplained in the original paper. For the replay methods, we use memory buffer 200 for MNIST and CIFAR-10 and 2000 for CIFAR-100 and Tiny-ImageNet as in [29, 34]. We use the hyper-parameters suggested by the authors. If we could not reproduce any result, we use 10% of the training data as a validation set to grid-search for good hyper-parameters. For our proposed methods, we report the hyper-parameters in Appendix G. All the results are averages over 5 runs with random seeds.

**Evaluation Metrics.**

**(1).** *Average classification accuracy* over all classes after learning the last task. The final class prediction depends *prediction methods* (see below). We also report *forgetting rate* in Appendix J.

**(2).** *Average AUC* (Area Under the ROC Curve) over all task models for the evaluation of OOD detection. AUC is the main measure used in OOD detection papers. Using this measure, we show that a better OOD detection method will result in a better CIL performance. Let $AUC_k$ be the AUC score of task $k$. It is computed by using only the model (or classes) of task $k$ to score the test data of task $k$ as the in-distribution (IND) data and the test data from other tasks as the out-of-distribution (OOD) data. The average AUC score is: $AUC = \sum_k AUC_k / n$, where $n$ is the number of tasks.

It is not straightforward to change existing CL algorithms to include a new OOD detection method that needs training, e.g., CSI, except for TIL (task incremental learning) methods, e.g., HAT and Sup. For HAT and Sup, we can simply switch their methods for learning each task with CSI (see Sec.4.4).

**Prediction Methods.** The theoretical result in Sec. 3 states that we use Eq. 2 to perform the final prediction. The first probability (WP) in Eq. 2 is easy to get as we can simply use the softmax values of the classes in each task. However, the second probability (TP) in Eq. 2 is tricky as each task is learned without the data of other tasks. There can be many options. We take the following approaches for prediction (which are a special case of Eq. 2, see below):

**(1).** For those approaches that use a single classification head to include all classes learned so far, we predict as follows (which is also the approach taken by the existing papers.)

$$\hat{y} = \arg\max f(x) \tag{8}$$

where $f(x)$ is the logit output of the network.

**(2).** For multi-head methods (e.g., HAT, HyperNet, and Sup), which use one head for each task, we use the concatenated output as

$$\hat{y} = \arg\max \bigoplus_k f(x)_k \tag{9}$$

where $\bigoplus$ indicate concatenation and $f(x)_k$ is the output of task $k$.[6]

These methods (in fact, they are the same method used in two different settings) is a special case of Eq. 2 if we define $OOD_k$ as $\sigma(\max f(x)_k)$, where $\sigma$ is the sigmoid. Hence, the theoretical results in Sec. 3 are still applicable. We present a detailed explanation about this prediction method and some other options in Appendix C. These two approaches work quite well.

---

[6]The Sup paper proposed an one-shot task-id prediction assuming that the test instances come in a batch and all belong to the same task like iTAML. We assume a single test instance per batch. Its task-id prediction results in accuracy of 50.2 on C10-5T, which is much lower than 62.6 by using Eq. 9. The task-id prediction of HyperNet also works poorly. The accuracy by its id prediction is 49.34 on C10-5T while it is 53.4 using Eq. 9. PR uses entropy to find task-id. Among many variations of PR, we use the variations that perform the best for each dataset with exemplar-free and single sample per batch at testing (i.e., no PR-BW).

### 4.3 Better OOD Detection Produces Better CIL Performance

The key theoretical result in Sec. 3 is that better OOD detection will produce better CIL performance. Recall our considered methods ODIN and CSI can perform both WP and OOD detection.

**Applying ODIN.** We first train the baseline models using their original algorithms, and then apply temperature scaling and input noise of ODIN at testing for each task (no training data needed). More precisely, the output of class $j$ in task $k$ changes by temperature scaling factor $\tau_k$ of task $k$ as

$$s(x; \tau_k)_j = e^{f(x)_{kj}/\tau_k} / \sum_j e^{f(x)_{kj}/\tau_k} \tag{10}$$

and the input changes by the noise factor $\epsilon_k$ as

$$\tilde{x} = x - \epsilon_k \text{sign}(-\nabla_x \log s(x; \tau_k)_{\hat{y}}) \tag{11}$$

where $\hat{y}$ is the class with the maximum output value in task $k$. This is a positive adversarial example inspired by [75]. The values $\tau_k$ and $\epsilon_k$ are hyper-parameters and we use the same values for all tasks except for PASS, for which we had to use a validation set to tune $\tau_k$ (see Appendix B).

Tab. 1 gives the results for C100-10T. The CIL results clearly show that the CIL performance increases if the AUC increases with ODIN. For instance, the CIL of DER++ and Sup improves from 53.71 to 55.29 and 44.58 to 46.74, respectively, as the AUC increases from 85.99 to 88.21 and 79.16 to 80.58. It shows that when this method is incorporated into each task model in existing trained CIL network, the CIL performance of the original method improves. We note that ODIN does not always improve the average AUC. For those experienced a decrease in AUC, the CIL performance also decreases except LwF. The inconsistency of LwF is due to its severe classification bias towards later tasks as discussed in BiC [32]. The temperature scaling in ODIN has a similar effect as the bias correction in BiC, and the CIL of LwF becomes close to that of BiC after the correction. Regardless of whether ODIN improves AUC or not, the positive correlation between AUC and CIL (except LwF) verifies the efficacy of Theorem 3, indicating better OOD detection results in better CIL performances.

**Applying CSI.** We now apply the OOD detection method CSI. Due to its sophisticated data augmentation, supervised constrative learning and results ensemble, it is hard to apply CSI to other baselines without fundamentally change them except for HAT and Sup (SupSup) as these methods are parameter isolation-based TIL methods. We can simply replace their model for training each task with CSI wholesale (the full detail is given in Appendix D). As mentioned earlier, both HAT and SupSup as TIL methods have almost no forgetting.

Table 1: Performance comparison based on C100-10T between the original output and the output post-processed with OOD detection technique ODIN. Note that ODIN is not applicable to iCaRL and Mnemonics as they are not based on softmax but some distance functions. The results for other datasets are reported in Appendix B.

| Method | OOD | AUC | CIL |
|---|---|---|---|
| OWM | Original | 71.31 | 28.91 |
| | ODIN | 70.06 | 28.88 |
| MUC | Original | 72.69 | 30.42 |
| | ODIN | 72.53 | 29.79 |
| PASS | Original | 69.89 | 33.00 |
| | ODIN | 69.60 | 31.00 |
| LwF | Original | 88.30 | 45.26 |
| | ODIN | 87.11 | 51.82 |
| BiC | Original | 87.89 | 52.92 |
| | ODIN | 86.73 | 48.65 |
| DER++ | Original | 85.99 | 53.71 |
| | ODIN | 88.21 | 55.29 |
| HAT | Original | 77.72 | 41.06 |
| | ODIN | 77.80 | 41.21 |
| HyperNet | Original | 71.82 | 30.23 |
| | ODIN | 72.32 | 30.83 |
| Sup | Original | 79.16 | 44.58 |
| | ODIN | 80.58 | 46.74 |

Tab. 2 reports the results of using CSI and ODIN. ODIN is a weaker OOD method than CSI. Both HAT and Sup improve greatly as the systems are equipped with a better OOD detection method CSI. These experiment results empirically demonstrate the efficacy of Theorem 3, i.e., the CIL performance can be improved if a better OOD detection method is used.

### 4.4 Full Comparison of HAT+CSI and Sup+CSI with Baselines

We now make a full comparison of the two strong systems (HAT+CSI and Sup+CSI) designed based on the theoretical results. These combinations are particularly attractive because both HAT and Sup are TIL systems and have little or no CF. Then a strong OOD method (that can also perform WP (within-task/IND prediction) will result in a strong CIL method. Since HAT and Sup are exemplar-free CL methods, HAT+CSI and Sup+CSI also do not need to save any previous task data. Tab. 3

Table 2: Average CIL and AUC of HAT and Sup with OOD detection methods ODIN and CSI. ODIN is a traditional OOD detection method while CSI is a recent OOD detection method known to be better than ODIN. As CL methods produce better OOD detection performance by CSI, their CIL performances are better than the ODIN counterparts.

| CL | OOD | C10-5T | | C100-10T | | C100-20T | | T-5T | | T-10T | |
|----|-----|--------|-----|----------|-----|----------|-----|------|-----|-------|-----|
| | | AUC | CIL | AUC | CIL | AUC | CIL | AUC | CIL | AUC | CIL |
| HAT | ODIN | 82.5 | 62.6 | 77.8 | 41.2 | 75.4 | 25.8 | 72.3 | 38.6 | 71.8 | 30.0 |
| | CSI | 91.2 | 87.8 | 84.5 | 63.3 | 86.5 | 54.6 | 76.5 | 45.7 | 78.5 | 47.1 |
| Sup | ODIN | 82.4 | 62.6 | 80.6 | 46.7 | 81.6 | 36.4 | 74.0 | 41.1 | 74.6 | 36.5 |
| | CSI | 91.6 | 86.0 | 86.8 | 65.1 | 88.3 | 60.2 | 77.1 | 48.9 | 79.4 | 45.7 |

Table 3: Average accuracy after all tasks are learned. Exemplar-free methods are italicized. † indicates that in their original papers, PASS and Mnemonics are pre-trained with the first half of the classes. Their results with pre-train are 50.1 and 53.5 on C100-10T, respectively, which are still much lower than the proposed HAT+CSI and Sup+CSI without pre-training. We do not use pre-training in our experiment for fairness. ∗ indicates that iCaRL and Mnemonics report average incremental accuracy in their original papers. We report average accuracy over all classes after all tasks are learned.

| Method | M-5T | C10-5T | C100-10T | C100-20T | T-5T | T-10T |
|--------|------|--------|----------|----------|------|-------|
| *OWM* | 95.8±0.13 | 51.8±0.05 | 28.9±0.60 | 24.1±0.26 | 10.0±0.55 | 8.6±0.42 |
| *MUC* | 74.9±0.46 | 52.9±1.03 | 30.4±1.18 | 14.2±0.30 | 33.6±0.19 | 17.4±0.17 |
| *PASS†* | 76.6±1.67 | 47.3±0.98 | 33.0±0.58 | 25.0±0.69 | 28.4±0.51 | 19.1±0.46 |
| LwF | 85.5±3.11 | 54.7±1.18 | 45.3±0.75 | 44.3±0.46 | 32.2±0.50 | 24.3±0.26 |
| iCaRL∗ | 96.0±0.43 | 63.4±1.11 | 51.4±0.99 | 47.8±0.48 | 37.0±0.41 | 28.3±0.18 |
| Mnemonics†∗ | 96.3±0.36 | 64.1±1.47 | 51.0±0.34 | 47.6±0.74 | 37.1±0.46 | 28.5±0.72 |
| BiC | 94.1±0.65 | 61.4±1.74 | 52.9±0.64 | 48.9±0.54 | 41.7±0.74 | 33.8±0.40 |
| DER++ | 95.3±0.69 | 66.0±1.20 | 53.7±1.30 | 46.6±1.44 | 35.8±0.77 | 30.5±0.47 |
| Co²L | | 65.6 | | | | |
| *CCG* | 97.3 | 70.1 | | | | |
| *HAT* | 81.9±3.74 | 62.7±1.45 | 41.1±0.93 | 25.6±0.51 | 38.5±1.85 | 29.8±0.65 |
| *HyperNet* | 56.6±4.85 | 53.4±2.19 | 30.2±1.54 | 18.7±1.10 | 7.9±0.69 | 5.3±0.50 |
| *Sup* | 70.1±1.51 | 62.4±1.45 | 44.6±0.44 | 34.7±0.30 | 41.8±1.50 | 36.5±0.36 |
| *PR-Ent* | 74.1 | 61.9 | 45.2 | | | |
| *HAT+CSI* | 94.4±0.26 | 87.8±0.71 | 63.3±1.00 | 54.6±0.92 | 45.7±0.26 | 47.1±0.18 |
| *Sup+CSI* | 80.7±2.71 | 86.0±0.41 | 65.1±0.39 | 60.2±0.51 | 48.9±0.25 | 45.7±0.76 |
| HAT+CSI+c | 96.9±0.30 | 88.0±0.48 | 65.2±0.71 | 58.0±0.45 | 51.7±0.37 | 47.6±0.32 |
| Sup+CSI+c | 81.0±2.30 | 87.3±0.37 | 65.2±0.37 | 60.5±0.64 | 49.2±0.28 | 46.2±0.53 |

shows that HAT and Sup equipped with CSI outperform the baselines by large margins. DER++, the best replay method, achieves 66.0 and 53.7 on C10-5T and C100-10T, respectively, while HAT+CSI achieves 87.8 and 63.3 and Sup+CSI achieves 86.0 and 65.1. The large performance gap remains consistent in more challenging problems, T-5T and T-10T. We note that Sup works very poorly on M-5T, but Sup+CSI improved it drastically, although still very weak compared to HAT+CSI.

Due to the definition of OOD in the prediction method and the fact that each task is trained separately in HAT and Sup, the outputs $f(x)_k$ from different tasks can be in different scales, which will result in incorrect predictions. To deal with the problem, we can calibrate the output as $\alpha_k f(x)_k + \beta_k$ and use $OOD_k = \sigma(\alpha_k f(x)_k + \beta_k)$. The optimal $\alpha_k^*$ and $\beta_k^*$ for each task $k$ can be found by optimization with a memory buffer to save a very small number of training examples from previous tasks like that in the replay-based methods. We refer the calibrated methods as HAT+CSI+c and Sup+CSI+c. They are trained by using the memory buffer of the same size as the replay methods (see Sec. 4.2). Tab. 3 shows that the calibration improves from their memory free versions, i.e., without calibration. We provide the details about how to train the calibration parameters $\alpha_k$ and $\beta_k$ in Appendix E.

As shown in Theorem 1, the CIL performance also depends on the TIL (WP) performance. We compare the TIL accuracies of the baselines and our methods in Tab. 4. Our systems again outperform the baselines by large margins on more challenging datasets (e.g., CIFAR100 and Tiny-ImageNet).

Table 4: TIL (WP) results of 3 best performing baselines and our methods. The full results are given in Appendix F. The calibrated versions (+c) of our methods are omitted as calibration does not affect TIL performances.

| Method | M-5T | C10-5T | C100-10T | C100-20T | T-5T | T-10T |
|---|---|---|---|---|---|---|
| DER++ | 99.7±0.08 | 92.0±0.54 | 84.0±9.43 | 86.6±9.44 | 57.4±1.31 | 60.0±0.74 |
| HAT | 99.9±0.02 | 96.7±0.18 | 84.0±0.23 | 85.0±0.98 | 61.2±0.72 | 63.8±0.41 |
| Sup | 99.6±0.01 | 96.6±0.21 | 87.9±0.27 | 91.6±0.15 | 64.3±0.24 | 68.4±0.22 |
| HAT+CSI | 99.9±0.00 | 98.7±0.06 | 92.0±0.37 | 94.3±0.06 | 68.4±0.16 | 72.4±0.21 |
| Sup+CSI | 99.0±0.08 | 98.7±0.07 | 93.0±0.13 | 95.3±0.20 | 65.9±0.25 | 74.1±0.28 |

### 4.5 Implications for Existing CL Methods, Open-World Learning and Future Research

**Implication for regularization and replay methods.** Regularization-based (exemplar-free) methods try to protect important parameters of old tasks to mitigate CF. However, since the training of each task does not consider OOD detection, TP will be weak, which causes difficulty for *inter-task class separation* (ICS) and thus low CL accuracy. Replay-based methods are better as the replay data from old tasks can be naturally regarded as OOD data for the current task, then a better OOD model is built, which improves TP. However, since the replay data is small. the OOD model is sub-optimal, especially for earlier tasks as their training cannot see any future task data. For both approaches, it will be beneficial to consider CF and OOD together in learning each task (e.g., [76]).

**Implication for open-world learning.** Since our theory says that CL needs OOD detection, and OOD detection is also the first step in open-world learning (OWL), CL and OWL are unified to form the *self-motivated open-world continual learning* [5] for learning or AI autonomy. That is, the AI agent can continually discover new tasks (OOD detection) and incrementally learn the tasks (CL) (see [76]) all on its own with no involvement of human engineers [5]. Further afield, this work is also related to curiosity-driven self-supervised learning [77] in reinforcement learning and 3D navigation.

**Limitation and future work.** The proposed theory provides a principled guidance on what needs to be done in order to achieve good CIL results, but it gives no guidance on how to do it. Although two example techniques are presented and evaluated, they are empirical. There are many options to define WP and TP (or OOD). An idea in [40] may be helpful in this regard. [40] argues that a continual learner should learn *holistic feature representations* of the input data, meaning to learn as many features as possible from the input data. The rationale is that if the system can learn all possible features from each task, then a future task does not have to learn those shared/intersecting features by modifying the parameters, which will result in less CF and also better ICS. A full representation of the IND data also improves OOD detection because the OOD score of a data point is basically the distance between the data point and the IND distribution. Only capturing a subset of features (e.g., by cross entropy) will result in poor OOD detection [78] because those missing features may be necessary to separate IND and some OOD data. In our future work, we will study how to optimize WP and TP/OOD and find the necessary conditions for them to do well.

## 5 Conclusion

This paper proposed a theoretical study on how to solve the highly challenging continual learning (CL) problem. *class incremental learning* (CIL) (the other popular CL setting is *task incremental learning* (TIL)). The theoretical result provides a principled guidance for designing better CIL algorithms. The paper first decomposed the CIL prediction into *within-task prediction* (WP) and *task-id prediction* (TP). WP is basically TIL. The paper further theoretically demonstrated that TP is correlated with *out-of-distribution* (OOD) detection. It then proved that a good performance of the two is both necessary and sufficient for good CIL performances. Based on the theoretical result, several new CIL methods were designed. They outperform strong baselines in CIL and also in TIL by a large margin. Finally, we also discussed the implications for existing CL techniques and open-world learning.

## Acknowledgments

The work of Gyuhak Kim, Zixuan Ke and Bing Liu was supported in part by a research contract from KDDI, two NSF grants (IIS-1910424 and IIS-1838770), and a DARPA contract HR001120C0023.

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
