# A Theoretical Study on Solving Continual Learning - Appendix

**Gyuhak Kim**[*1], **Changnan Xiao**[*2], **Tatsuya Konishi**[†3], **Zixuan Ke**[1], **Bing Liu**[‡1]
[1] University of Illinois at Chicago
[2] ByteDance
[3] KDDI Research

## A  Proof of Theorems and Corollaries

### A.1  Proof of Theorem 1

*Proof.* Since

$$H_{CIL}(x) = H(y, \{\mathbf{P}(x \in \mathbf{X}_{k,j}|D)\}_{k,j})$$
$$= -\sum_{k,j} y_{k,j} \log \mathbf{P}(x \in \mathbf{X}_{k,j}|D)$$
$$= -\log \mathbf{P}(x \in \mathbf{X}_{k_0,j_0}|D),$$
$$H_{WP}(x) = H(\tilde{y}, \{\mathbf{P}(x \in \mathbf{X}_{k_0,j}|x \in \mathbf{X}_{k_0}, D)\}_j)$$
$$= -\sum_j y_{k_0,j} \log \mathbf{P}(x \in \mathbf{X}_{k_0,j}|x \in \mathbf{X}_{k_0}, D)$$
$$= -\log \mathbf{P}(x \in \mathbf{X}_{k_0,j_0}|x \in \mathbf{X}_{k_0}, D),$$

and

$$H_{TP}(x) = H(\bar{y}, \{\mathbf{P}(x \in \mathbf{X}_k|D)\}_k)$$
$$= -\sum_k \bar{y}_k \log \mathbf{P}(x \in \mathbf{X}_k|D)$$
$$= -\log \mathbf{P}(x \in \mathbf{X}_{k_0}|D),$$

we have

$$H_{CIL}(x) = -\log \mathbf{P}(x \in \mathbf{X}_{k_0,j_0}|D)$$
$$= -\log \mathbf{P}(x \in \mathbf{X}_{k_0,j_0}|x \in \mathbf{X}_{k_0}, D) - \log \mathbf{P}(x \in \mathbf{X}_{k_0}|D)$$
$$= H_{WP}(x) + H_{TP}(x)$$
$$\leq \epsilon + \delta.$$

$\square$

### A.2  Proof of Corollary 1.

*Proof.* By proof of Theorem 1, we have

$$H_{CIL}(x) = H_{WP}(x) + H_{TP}(x).$$

Taking expectations on both sides, we have i)

$$\mathbb{E}_{x \sim U(\mathbf{X})}[H_{CIL}(x)] = \mathbb{E}_{x \sim U(\mathbf{X})}[H_{WP}(x)] + \mathbb{E}_{x \sim U(\mathbf{X})}[H_{TP}(x)]$$
$$\leq \mathbb{E}_{x \sim U(\mathbf{X})}[H_{WP}(x)] + \delta.$$

---

[*]Equal contribution
[†]The work was done when this author was visiting Bing Liu's group at University of Illinois at Chicago
[‡]Correspondance author. Bing Liu <liub@uic.edu>

36th Conference on Neural Information Processing Systems (NeurIPS 2022).

and ii)
$$\mathbb{E}_{x \sim U(\mathbf{X})}[H_{CIL}(x)] = \mathbb{E}_{x \sim U(\mathbf{X})}[H_{WP}(x)] + \mathbb{E}_{x \sim U(\mathbf{X})}[H_{TP}(x)]$$
$$\leq \epsilon + \mathbb{E}_{x \sim U(\mathbf{X})}[H_{TP}(x)].$$

$\square$

## A.3 Proof of Theorem 2.

*Proof.* i) Assume $x \in \mathbf{X}_{k_0}$.

For $k = k_0$, we have
$$H_{OOD,k_0}(x) = -\log \mathbf{P}'_{k_0}(x \in \mathbf{X}_{k_0}|D)$$
$$= -\log \mathbf{P}(x \in \mathbf{X}_{k_0}|D)$$
$$= H_{TP}(x) \leq \delta.$$

For $k \neq k_0$, we have
$$H_{OOD,k}(x) = -\log \mathbf{P}'_k(x \notin \mathbf{X}_k|D)$$
$$= -\log(1 - \mathbf{P}'_k(x \in \mathbf{X}_k|D))$$
$$= -\log(1 - \mathbf{P}(x \in \mathbf{X}_k|D))$$
$$= -\log \mathbf{P}(x \in \cup_{k' \neq k}\mathbf{X}_{k'}|D)$$
$$\leq -\log \mathbf{P}(x \in \mathbf{X}_{k_0}|D)$$
$$= H_{TP}(x) \leq \delta.$$

ii) Assume $x \in \mathbf{X}_{k_0}$.

For $k = k_0$, by $H_{OOD,k_0}(x) \leq \delta_{k_0}$, we have
$$-\log \mathbf{P}'_{k_0}(x \in \mathbf{X}_{k_0}|D) \leq \delta_{k_0},$$
which means
$$\mathbf{P}'_{k_0}(x \in \mathbf{X}_{k_0}|D) \geq e^{-\delta_{k_0}}.$$
For $k \neq k_0$, by $H_{OOD,k}(x) \leq \delta_k$, we have
$$-\log \mathbf{P}'_k(x \notin \mathbf{X}_k|D) \leq \delta_k,$$
which means
$$\mathbf{P}'_k(x \in \mathbf{X}_k|D) \leq 1 - e^{-\delta_k}.$$

Therefore, we have
$$\mathbf{P}(x \in \mathbf{X}_{k_0}|D) = \frac{\mathbf{P}'_{k_0}(x \in \mathbf{X}_{k_0}|D)}{\sum_{k'} \mathbf{P}'_{k'}(x \in \mathbf{X}_{k'}|D)}$$
$$\geq \frac{e^{-\delta_{k_0}}}{1 + \sum_{k \neq k_0} 1 - e^{-\delta_k}}$$
$$= \frac{e^{-\delta_{k_0}}}{e^{-\delta_{k_0}} + \sum_k 1 - e^{-\delta_k}}$$
$$= \frac{1}{1 + e^{\delta_{k_0}} \sum_k 1 - e^{-\delta_k}}.$$

Hence,
$$H_{TP}(x) = -\log \mathbf{P}(x \in \mathbf{X}_{k_0}|D)$$
$$\leq -\log \frac{1}{1 + e^{\delta_{k_0}} \sum_k 1 - e^{-\delta_k}}$$
$$= \log[1 + e^{\delta_{k_0}} \sum_k 1 - e^{-\delta_k}]$$
$$\leq e^{\delta_{k_0}}(\sum_k 1 - e^{-\delta_k})$$
$$= (\sum_k \mathbf{1}_{x \in \mathbf{X}_k} e^{\delta_k})(\sum_k 1 - e^{-\delta_k}).$$

$\square$

## A.4 Proof of Theorem 3.

*Proof.* Using Theorem 1 and 2,

$$
\begin{aligned}
H_{CIL}(x) &= -\log \mathbf{P}(x \in \mathbf{X}_{k_0,j_0}|D) \\
&= -\log \mathbf{P}(x \in \mathbf{X}_{k_0,j_0}|x \in \mathbf{X}_{k_0}, D) - \log \mathbf{P}(x \in \mathbf{X}_{k_0}|D) \\
&= H_{WP}(x) + H_{TP}(x) \\
&\leq \epsilon + H_{TP}(x) \\
&\leq \epsilon + (\sum_k \mathbf{1}_{x \in \mathbf{X}_k} e^{\delta_k})(\sum_k 1 - e^{-\delta_k})
\end{aligned}
$$

$\square$

## A.5 Proof of Theorem 4.

*Proof.* i) Assume $x \in \mathbf{X}_{k_0,j_0} \subset \mathbf{X}_{k_0}$.

Define $\mathbf{P}(x \in \mathbf{X}_{k,j}|x \in \mathbf{X}_k, D) = \mathbf{P}(x \in \mathbf{X}_{k,j}|D)$.

According to proof of Theorem 1,

$$
\begin{aligned}
H_{WP}(x) &= -\log \mathbf{P}(x \in \mathbf{X}_{k_0,j_0}|x \in \mathbf{X}_{k_0}, D), \\
H_{CIL}(x) &= -\log \mathbf{P}(x \in \mathbf{X}_{k_0,j_0}|D).
\end{aligned}
$$

Hence, we have

$$
\begin{aligned}
H_{WP}(x) &= -\log \mathbf{P}(x \in \mathbf{X}_{k_0,j_0}|x \in \mathbf{X}_{k_0}, D) \\
&= -\log \mathbf{P}(x \in \mathbf{X}_{k_0,j_0}|D) \\
&= H_{CIL}(x) \leq \eta.
\end{aligned}
$$

ii) Assume $x \in \mathbf{X}_{k_0,j_0} \subset \mathbf{X}_{k_0}$.

Define $\mathbf{P}(x \in \mathbf{X}_k|D) = \sum_j \mathbf{P}(x \in \mathbf{X}_{k,j}|D)$.

According to proof of Theorem 1,

$$
\begin{aligned}
H_{TP}(x) &= -\log \mathbf{P}(x \in \mathbf{X}_{k_0}|D), \\
H_{CIL}(x) &= -\log \mathbf{P}(x \in \mathbf{X}_{k_0,j_0}|D).
\end{aligned}
$$

Hence, we have

$$
\begin{aligned}
H_{TP}(x) &= -\log \mathbf{P}(x \in \mathbf{X}_{k_0}|D) \\
&= -\log \sum_j \mathbf{P}(x \in \mathbf{X}_{k_0,j}|D) \\
&\leq -\log \mathbf{P}(x \in \mathbf{X}_{k_0,j_0}|D) \\
&= H_{CIL}(x) \leq \eta.
\end{aligned}
$$

iii) Assume $x \in \mathbf{X}_{k_0,j_0} \subset \mathbf{X}_{k_0}$.

Define $\mathbf{P}'_i(x \in \mathbf{X}_k|D) = \mathbf{P}(x \in \mathbf{X}_k|D) = \sum_j \mathbf{P}(x \in \mathbf{X}_{k,j}|D)$.

According to proof of Theorem 4 ii), we have

$$
H_{TP}(x) \leq \eta.
$$

According to proof of Theorem 2 i), we have

$$
H_{OOD,i}(x) \leq H_{TP}(x).
$$

Therefore,

$$
H_{OOD,i}(x) \leq H_{TP}(x) \leq \eta.
$$

$\square$

Table 5: Performance comparison between the original output and output post-processed with OOD detection technique ODIN. Note that ODIN is not applicable to iCaRL and Mnemonics as they are not based on softmax but some distance functions. The result for C100-10T are reported in the main paper.

| Method | OOD | M-5T | | C10-5T | | C100-20T | | T-5T | | T-10T | |
|--------|-----|------|-----|--------|-----|----------|-----|------|-----|-------|-----|
| | | AUC | CIL | AUC | CIL | AUC | CIL | AUC | CIL | AUC | CIL |
| OWM | Original | 99.13 | 95.81 | 81.33 | 51.79 | 71.90 | 24.15 | 58.49 | 10.00 | 59.48 | 8.57 |
| | ODIN | 98.86 | 95.16 | 71.72 | 40.65 | 68.52 | 23.05 | 58.46 | 10.77 | 59.38 | 9.52 |
| MUC | Original | 92.27 | 74.90 | 79.49 | 52.85 | 66.20 | 14.19 | 68.42 | 33.57 | 62.63 | 17.39 |
| | ODIN | 92.67 | 75.71 | 79.54 | 53.22 | 65.72 | 14.11 | 68.32 | 33.45 | 62.17 | 17.27 |
| PASS | Original | 98.74 | 76.58 | 66.51 | 47.34 | 70.26 | 24.99 | 65.18 | 28.40 | 63.27 | 19.07 |
| | ODIN | 90.40 | 74.33 | 63.08 | 35.20 | 69.81 | 21.83 | 65.93 | 29.03 | 62.73 | 17.78 |
| LwF | Original | 99.19 | 85.46 | 89.39 | 54.67 | 89.84 | 44.33 | 78.20 | 32.17 | 79.43 | 24.28 |
| | ODIN | 98.52 | 90.39 | 88.94 | 63.04 | 88.68 | 47.56 | 76.83 | 36.20 | 77.02 | 28.29 |
| BiC | Original | 99.40 | 94.11 | 90.89 | 61.41 | 89.46 | 48.92 | 80.17 | 41.75 | 80.37 | 33.77 |
| | ODIN | 98.57 | 95.14 | 91.86 | 64.29 | 87.89 | 47.40 | 74.54 | 37.40 | 76.27 | 29.06 |
| DER++ | Original | 99.78 | 95.29 | 90.16 | 66.04 | 85.44 | 46.59 | 71.80 | 35.80 | 72.41 | 30.49 |
| | ODIN | 99.09 | 94.96 | 87.08 | 63.07 | 87.72 | 49.26 | 73.92 | 37.87 | 72.91 | 32.52 |
| HAT | Original | 94.46 | 81.86 | 82.47 | 62.67 | 75.35 | 25.64 | 72.28 | 38.46 | 71.82 | 29.78 |
| | ODIN | 94.56 | 82.06 | 82.45 | 62.60 | 75.36 | 25.84 | 72.31 | 38.61 | 71.83 | 30.01 |
| HyperNet | Original | 85.83 | 56.55 | 78.54 | 53.40 | 72.04 | 18.67 | 54.58 | 7.91 | 55.37 | 5.32 |
| | ODIN | 86.89 | 64.31 | 79.39 | 56.72 | 73.89 | 23.8 | 54.60 | 8.64 | 55.53 | 6.91 |
| Sup | Original | 90.70 | 70.06 | 79.16 | 62.37 | 81.14 | 34.70 | 74.13 | 41.82 | 74.59 | 36.46 |
| | ODIN | 90.68 | 69.70 | 82.38 | 62.63 | 81.48 | 36.35 | 73.96 | 41.10 | 74.61 | 36.46 |

## B    Additional Results and Explanation Regarding Table 1 in the Main Paper

In Sec. 4.3, we showed that a better OOD detection improves CIL performance. For the post-processing method ODIN, we only reported the results on C100-10T due to space limitations. Tab. 5 shows the results on the other datasets.

A continual learning method with a better AUC shows a better CIL performance than other methods with lower AUC. For instance, original HAT achieves AUC of 82.47 while HyperNet achieves 78.54 on C10-5T. The CIL for HAT is 62.67 while it is 53.40 for HyperNet. However, there are some exceptions that this comparison does not hold. An example is LwF. Its AUC and CIL are 89.39 and 54.67 on C10-5T. Although its AUC is better than HAT, the CIL is lower. This is due to the fact that CIL improves with WP and TP according to Theorem 1. The contraposition of Theorem 4 also says if the cross-entropy of TIL is large, that of CIL is also large. Indeed, the average within-task prediction (WP) accuracy for LwF on C10-5T is 95.2 while the same for HAT is 96.7. Improving WP is also important in achieving good CIL performances.

For PASS, we had to tune $\tau_k$ using a validation set. This is because the softmax in Eq. 10 improves AUC by making the IND (in-distribution) and OOD scores more separable within a task, but deteriorates the final scores across tasks. To be specific, the test instances are predicted as one of the classes in the first task after softmax because the relative values between classes in task 1 is larger than the other tasks in PASS. Therefore, larger $\tau_1$ and smaller $\tau_k$, for $k > 1$, are chosen to compensate the relative values.

## C    Definitions of TP

As noted in the main paper, the class prediction in Eq. 2 varies by definition of WP and TP. The precise definition of WP and TP depends on implementation. Due to this subjectivity, we follow the prediction method as the existing methods in continual learning, which is the $\arg\max$ over the output. In this section, we show that the $\arg\max$ over output is a special case of Eq. 2. We also provide CIL results using different definitions of TP.

We first establish another theorem. This is an extension of Theorem 2 and connects the standard prediction method to our analysis.

**Theorem 5** (Extension of Theorem 2). *i) If $H_{TP}(x) \leq \delta$, let $\mathbf{P}'_k(x \in \mathbf{X}_k|D) = \mathbf{P}(x \in \mathbf{X}_k|D)^{1/\tau_k}$, $\forall \tau_k > 0$, then $H_{OOD,k}(x) \leq \max(\delta/\tau_k, -\log(1 - (1 - e^{-\delta})^{1/\tau_k}), \forall k = 1, \ldots, T$.*

*ii) If $H_{OOD,k}(x) \leq \delta_k, k = 1, \ldots, T$, let $\mathbf{P}(x \in \mathbf{X}_k|D) = \frac{\mathbf{P}'_k(x \in \mathbf{X}_k|D)^{1/\tau_k}}{\sum_j \mathbf{P}'_j(x \in \mathbf{X}_j|D)^{1/\tau_j}}$, $\forall \tau_k > 0$, then*

$$H_{TP}(x) \leq \sum_k \frac{\mathbf{1}_{x \in \mathbf{X}_k} \delta_k}{\tau_k} + \frac{\sum_k (1 - e^{-\delta_k})^{1/\tau_k}}{\sum_k \mathbf{1}_{x \in \mathbf{X}_k} (1 - (1 - e^{-\delta_k})^{1/\tau_k})}, \text{ where } \mathbf{1}_{x \in \mathbf{X}_k} \text{ is an indicator function.}$$

In Theorem 5 (proof appears later), we can observe that $\delta/\tau_k$ decreases with the increase of $\tau_k$, while $-\log(1 - (1 - e^{-\delta})^{1/\tau_k})$ increases. Hence, when TP is given, let $\delta = H_{TP}(x)$, we can find the optimal $\tau_i$ to define OOD by solving $\delta/\tau_k = -\log(1 - (1 - e^{-\delta})^{1/\tau_k})$. Similarly, given OOD, let $\delta_k = H_{OOD,k}(x)$, we can find the optimal $\tau_1, \ldots, \tau_T$ to define TP by finding the global minima of $\sum_k \frac{\mathbf{1}_{x \in \mathbf{X}_k} \delta_k}{\tau_k} + \frac{\sum_k (1 - e^{-\delta_k})^{1/\tau_k}}{\sum_k \mathbf{1}_{x \in \mathbf{X}_k} (1 - (1 - e^{-\delta_k})^{1/\tau_k})}$. The optimal $\tau_k$ can be found using a memory buffer to save a small number of previous data like that in a replay-based continual learning method.

In Theorem 5 (ii), let $\mathbf{P}'_k(x \in \mathbf{X}_k|D) = \sigma(\max f(x)_k)$, where $\sigma$ is the sigmoid and $f(x)_k$ is the output of task $k$ and choose $\tau_k \approx 0$ for each $k$. Then $\mathbf{P}(x \in \mathbf{X}_k|D)$ becomes approximately 1 for the task $k$ where the maximum logit value appears and 0 for the rest tasks. Therefore, Eq. 2 in the paper

$$\mathbf{P}(x \in \mathbf{X}_{k,j}|D) = \mathbf{P}(x \in \mathbf{X}_{k,j}|x \in \mathbf{X}_k, D)\mathbf{P}(x \in \mathbf{X}_k|D)$$

is zero for all classes in tasks $k' \neq k$. Since only the probabilities of classes in task $k$ are non-zero, taking $\arg\max$ over all class probabilities gives the same class as $\arg\max$ over output logits.

We have also tried another definition of WP and TP. The considered WP is

$$\mathbf{P}(x \in \mathbf{X}_{k,j}|x \in \mathbf{X}_k, D) = \frac{e^{f(x)_{kj}/\nu_k}}{\sum_j e^{f(x)_{kj}/\nu_k}}, \tag{12}$$

where $\nu_k$ is a temperature scaling parameter for task $k$, and the TP is

$$\mathbf{P}(x \in \mathbf{X}_k|D) = \frac{\mathbf{P}'_k(x \in \mathbf{X}_k|D)}{\sum_k \mathbf{P}'_k(x \in \mathbf{X}_k|D)}, \tag{13}$$

where $\mathbf{P}'_k(x \in \mathbf{X}_k|D) = \max_j e^{f(x)_{kj}/\tau_k}/\sum_j e^{f(x)_{kj}/\tau_k}$ and $\tau_k$ is a temperature scaling parameter. This is the maximum softmax of task $k$. We choose $\nu_k = 0.1$ and $\tau_k = 5$ for all $k$. A good $\tau$ and $\nu$ can be found using grid search on a validation set. However, one can also find the optimal values by optimization using some past data saved for memory buffer. The CIL results for the new prediction method is in Tab. 6.

*Proof of Theorem 5.* i) Assume $x \in \mathbf{X}_{k_0}$.

For $k = k_0$, we have

$$\begin{aligned}
H_{OOD,k_0}(x) &= -\log \mathbf{P}'_{k_0}(x \in \mathbf{X}_{k_0}|D) \\
&= -\frac{1}{\tau_{k_0}} \log \mathbf{P}(x \in \mathbf{X}_{k_0}|D) \\
&= \frac{1}{\tau_{k_0}} H_{TP}(x) \leq \frac{\delta}{\tau_{k_0}}.
\end{aligned}$$

For $k \neq k_0$, we have

$$\begin{aligned}
H_{OOD,k}(x) &= -\log \mathbf{P}'_k(x \notin \mathbf{X}_k|D) \\
&= -\log(1 - \mathbf{P}'_k(x \in \mathbf{X}_k|D)) \\
&= -\log(1 - \mathbf{P}(x \in \mathbf{X}_k|D)^{1/\tau_k}) \\
&= -\log(1 - (1 - \mathbf{P}(x \in \cup_{k' \neq k}\mathbf{X}_{k'}|D))^{1/\tau_k}) \\
&\leq -\log(1 - (1 - \mathbf{P}(x \in \mathbf{X}_{k_0}|D))^{1/\tau_k}) \\
&= -\log(1 - (1 - e^{-H_{TP}(x)})^{1/\tau_k}) \\
&\leq -\log(1 - (1 - e^{-\delta})^{1/\tau_k}).
\end{aligned}$$

ii) Assume $x \in \mathbf{X}_{k_0}$.

Table 6: Average classification accuracy. The results are based on class prediction method defined with WP and TP in Eq. 12 and Eq. 13, respectively. The results can improve by finding optimal temperature scaling parameters.

| Method | M-5T | C10-5T | C100-10T | C100-20T | T-5T | T-10T |
|---|---|---|---|---|---|---|
| *OWM* | 95.1±0.11 | 40.6±0.47 | 28.6±0.82 | 22.9±0.32 | 10.4±0.54 | 9.2±0.35 |
| *MUC* | 75.7±0.51 | 53.2±1.32 | 30.6±1.21 | 14.0±0.12 | 33.1±0.18 | 17.2±0.13 |
| *PASS*[†] | 64.5±2.64 | 33.6±0.71 | 18.5±1.85 | 20.8±0.85 | 21.4±0.44 | 13.0±0.55 |
| LwF | 90.4±1.18 | 63.0±0.34 | 51.9±0.88 | 47.5±0.62 | 35.9±0.32 | 27.8±0.29 |
| iCaRL[*] | 87.4±4.89 | 65.3±0.83 | 52.9±0.39 | 48.2±0.70 | 34.8±0.34 | 27.3±0.17 |
| Mnemonics[†*] | 91.8±1.03 | 65.6±1.55 | 50.7±0.72 | 47.9±0.71 | 36.3±0.30 | 27.7±0.78 |
| BiC | 95.1±0.47 | 65.5±0.81 | 50.8±0.69 | 47.2±0.71 | 37.0±0.58 | 29.1±0.34 |
| DER++ | 94.9±0.50 | 63.1±1.12 | 54.6±1.21 | 48.9±1.18 | 37.4±0.72 | 32.1±0.44 |
| *HAT* | 82.1±3.77 | 62.6±1.31 | 41.5±0.80 | 25.9±0.56 | 38.9±1.62 | 30.1±0.52 |
| *HyperNet* | 64.3±2.98 | 56.7±1.23 | 32.4±1.07 | 24.5±1.12 | 8.9±0.58 | 7.0±0.52 |
| *Sup* | 69.7±0.97 | 62.6±1.11 | 46.8±0.34 | 36.0±0.32 | 41.5±1.17 | 35.7±0.40 |
| *HAT+CSI* | 88.7±1.27 | 85.2±0.92 | 62.9±1.07 | 53.6±0.84 | 47.0±0.38 | 46.2±0.30 |
| *Sup+CSI* | 64.9±1.95 | 87.4±0.40 | 66.6±0.23 | 60.5±0.89 | 47.7±0.30 | 46.3±0.30 |
| HAT+CSI+c | 93.4±0.43 | 85.2±0.94 | 63.6±0.69 | 55.4±0.79 | 51.4±0.38 | 46.5±0.26 |
| Sup+CSI+c | 62.2±3.49 | 86.2±0.79 | 67.0±0.14 | 60.4±1.04 | 48.2±0.35 | 46.1±0.32 |

For $k = k_0$, by $H_{OOD,k_0}(x) \leq \delta_{k_0}$, we have

$$-\log \mathbf{P}'_{k_0}(x \in \mathbf{X}_{k_0}|D) \leq \delta_{k_0},$$

which means

$$\mathbf{P}'_{k_0}(x \in \mathbf{X}_{k_0}|D) \geq e^{-\delta_{k_0}}.$$

For $k \neq k_0$, by $H_{OOD,k}(x) \leq \delta_k$, we have

$$-\log \mathbf{P}'_k(x \notin \mathbf{X}_k|D) \leq \delta_k,$$

which means

$$\mathbf{P}'_k(x \in \mathbf{X}_k|D) \leq 1 - e^{-\delta_k}.$$

Therefore, we have

$$
\begin{aligned}
\mathbf{P}(x \in \mathbf{X}_{k_0}|D) &= \frac{\mathbf{P}'_{k_0}(x \in \mathbf{X}_{k_0}|D)^{1/\tau_{k_0}}}{\sum_k \mathbf{P}'_k(x \in \mathbf{X}_k|D)^{1/\tau_k}} \\
&\geq \frac{e^{-\delta_{k_0}/\tau_{k_0}}}{1 + \sum_{k \neq k_0}(1 - e^{-\delta_k})^{1/\tau_k}} \\
&= \frac{e^{-\delta_{k_0}/\tau_{k_0}}}{1 - (1 - e^{-\delta_{k_0}})^{1/\tau_{k_0}} + \sum_k(1 - e^{-\delta_k})^{1/\tau_k}} \\
&= \frac{e^{-\delta_{k_0}/\tau_{k_0}}}{1 - (1 - e^{-\delta_{k_0}})^{1/\tau_{k_0}}} \cdot \frac{1}{1 + \frac{\sum_k(1 - e^{-\delta_k})^{1/\tau_k}}{1 - (1 - e^{-\delta_{k_0}})^{1/\tau_{k_0}}}}.
\end{aligned}
$$

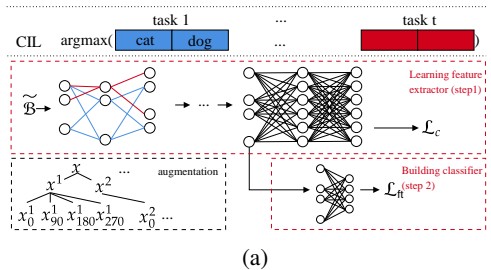 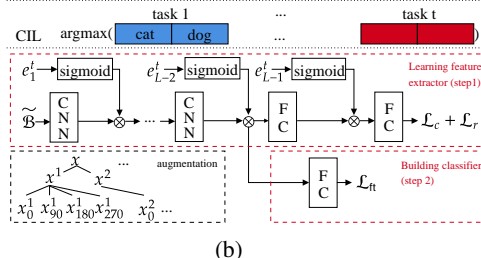

(a)                              (b)

Figure 1: Overview of prediction and training framework of Sup+CSI and HAT+CSI. (a) Sup+CSI: The CIL prediction is made by taking argmax over the concatenated output values from each task. In training the network, the training batch is augmented to give different views of samples for contrastive training in the OOD detection algorithm CSI. The training consists of two steps following CSI. The first step is learning the feature extractor. In this step, the Edge Popup algorithm [4] is applied to find a sparse network for each task. The sparse networks, which are indicated by edges of different colors in the diagram. The second step fine-tunes the classifier only the fixed feature extractor. (b) HAT+CSI: The CIL prediction is also made by argmax over the concatenated output from each task as Sup+CSI method. Due to the OOD detection algorithm CSI, the overall training process is similar to Sup+CSI except that it applies the hard attention algorithm [1]. In training feature extractor, task embeddings are applied to find hard masks at each layer. Then given the learned feature representations, fine-tunes the classifier in step 2.

Hence,

$$
\begin{aligned}
H_{TP}(x) &= -\log \mathbf{P}(x \in \mathbf{X}_{k_0}|D) \\
&\leq -\log \frac{e^{-\delta_{k_0}/\tau_{k_0}}}{1-(1-e^{-\delta_{k_0}})^{1/\tau_{k_0}}} \cdot \frac{1}{1+\frac{\sum_k (1-e^{-\delta_k})^{1/\tau_k}}{1-(1-e^{-\delta_{k_0}})^{1/\tau_{k_0}}}} \\
&= \frac{\delta_{k_0}}{\tau_{k_0}} + \log[1-(1-e^{-\delta_{k_0}})^{1/\tau_{k_0}}] + \log\left[1+\frac{\sum_k (1-e^{-\delta_k})^{1/\tau_k}}{1-(1-e^{-\delta_{k_0}})^{1/\tau_{k_0}}}\right] \\
&\leq \frac{\delta_{k_0}}{\tau_{k_0}} + \frac{\sum_k (1-e^{-\delta_k})^{1/\tau_k}}{1-(1-e^{-\delta_{k_0}})^{1/\tau_{k_0}}} \\
&= \sum_k \frac{\mathbf{1}_{x \in \mathbf{X}_k}\delta_k}{\tau_k} + \frac{\sum_k (1-e^{-\delta_k})^{1/\tau_k}}{\sum_k \mathbf{1}_{x \in \mathbf{X}_k}(1-(1-e^{-\delta_k})^{1/\tau_k})}.
\end{aligned}
$$

$\square$

# D   Details of HAT, Sup, and CSI

We have proposed two highly effective new CIL methods, HAT+CSI and Sup+CSI, by integrating the existing parameter isolation based continual learning (CL) method HAT [1] or Sup [2] with the strong OOD detection method CSI [3]. We replaced the training loss of HAT and Sup by that of CSI while applying the continual learning techniques of the respective method. In this section, we overview Sup, HAT, and CSI, and explain how to train them continually. Figure 1 shows the overall training frameworks of Sup+CSI and HAT+CSI.

Denote feature extractor by $h$, classifier by $f$, and the parameters by $\mathbf{W}$. In the main paper, we denote the output of task $k$ by $f(x)_k$ for both a single-head or multi-head method (e.g., Eq. 9) for consistency. In this section, we use $f(x,k)$ to indicate the output of task $k$ to be more explicit as both HAT and Sup are multi-head methods (one head for each task) designed for task incremental learning (TIL).

## D.1 Sup

SupSup (Sup) [2] trains supermasks by Edge Popup algorithm [4]. More precisely, given initial $\mathbf{W}$, find binary masks $\mathbf{M}_k$ for task $k$ to minimize the cross-entropy loss

$$\mathcal{L} = -\frac{1}{|\mathbf{X}_k|} \sum \log p(y|x, k), \tag{14}$$

where $\mathbf{X}_k$ is the training data for task $k$, and

$$p(y|x, k) = f(h(x; \mathbf{W} \otimes \mathbf{M}_k)), \tag{15}$$

where $\otimes$ indicates an element-wise product. The masks are obtained by selecting the top $p\%$ of entries in the score matrices $\mathbf{V}$. The $p$ value determines the sparsity of the mask $\mathbf{M}_k$. The subnetwork found by Edge Popup algorithm is indicated by different colors in Figure 1(a).

Given the task-id $k$ of a test instance at inference, the system (which is referred as Sup GG in the original Sup paper) uses the task-specific mask $\mathbf{M}_k$ to obtain the classification output. By integrating the OOD detection method, CSI, during training, Sup+CSI does not require to know the task-id of test instance, which makes Sup+CSI applicable to CIL (class incremental learning).

## D.2 HAT

We now discuss the hard attention (mask) mechanism of HAT [1]. It finds binary masks $a_l^k$ for each layer $l$ and task $k$, and uses them to block/unblock information flow at forward and backward pass. More precisely, the hard attention is defined as

$$a_l^k = \sigma(se_l^k), \tag{16}$$

where $\sigma$ is the sigmoid, $s$ is a positive constant, and $e_l^k$ is a learnable embedding. To approximate the binary mask, the system uses a large $s$ value. The attention is applied to the output at each layer as

$$h_l' = a_l^k \otimes h_l, \tag{17}$$

where $\otimes$ is an element-wise product, and

$$h_l = \text{ReLU}(W_l h_{l-1} + b_l). \tag{18}$$

The neurons with attention value 1 is important for task $k$ while those with zero attention value are not necessary for the task, and thus they can be freely changed without affecting the output value $h_l'$. The system needs to know which neurons are important to protect the previous knowledge from forgetting. Denote the accumulated attentions of all previous tasks by

$$a_l^{<k} = \max(a_l^{<k-1}, a_l^{k-1}), \tag{19}$$

where $a_l^0$ is the zero vector and $\max$ is an element-wise maximum. The gradients of parameters corresponding to important neurons is modified as

$$\nabla w_{ij,l}' = \left(1 - \min\left(a_{i,l}^{<k}, a_{j,l-1}^{<k}\right)\right) \nabla w_{ij,l}, \tag{20}$$

where $a_{i,l}^{<k}$ is the $i$'th unit of $a_l^{<k}$ and $l = 1, \cdots, L-1$. The hard attention is not applied to the last layer $L$ since it is a task-specific classification layer.

To encourage sparsity in $a_l^k$, the system uses regularization as

$$\mathcal{L}_r = \lambda_k \frac{\sum_l \sum_i a_{i,l}^k \left(1 - a_{i,l}^{<k}\right)}{\sum_l \sum_i \left(1 - a_{i,l}^{<k}\right)}, \tag{21}$$

where $\lambda_k$ is a hyper-parameter. The system minimizes the loss

$$\mathcal{L} = \mathcal{L}_{ce} + \mathcal{L}_r, \tag{22}$$

where $\mathcal{L}_{ce}$ is the cross-entropy loss. The overall framework of the algorithm is shown in Figure 1(b).

## D.3 CSI

We now explain the OOD detection method CSI, and how to incorporate it in HAT and Sup. CSI is based on contrastive learning [5, 6] and data augmentation due to their excellent performance [3]. Since this section focuses on how to learn a single task based on OOD detection, we omit the task-id unless necessary. The OOD training process is similar to that of contrastive learning. It consists of two steps: 1) learning the feature representation by the composite $g \circ h$, where $h$ is a feature extractor and $g$ is a projection to contrastive representation, and 2) learning a linear classifier $f$ mapping the feature representation of $h$ to the label space. This two step training process is outlined in Figure 1(a) and (b). In the following, we describe the training process: contrastive learning for feature representation learning (1), and OOD classifier building (2). We then explain how to make a prediction based on an ensemble method to further improve prediction.

### D.3.1 Contrastive Loss for Feature Learning.

This is step 1. Supervised contrastive learning is used to try to repel data of different classes and align data of the same class more closely to make it easier to classify them. A key operation is data augmentation via transformations.

Given a batch of $N$ samples, each sample $x$ is first duplicated and each version then goes through *three initial augmentations* (horizontal flip, color changes, and Inception crop [7]) to generate two different views $x^1$ and $x^2$ (they keep the same class label as $x$). Denote the augmented batch by $\mathcal{B}$, which now has $2N$ samples. In [8, 3], it was shown that using image rotations is effective in learning OOD detection models because such rotations can effectively serve as out-of-distribution (OOD) training data. For each augmented sample $x \in \mathcal{B}$ with class $y$ of a task, we rotate $x$ by $90°, 180°, 270°$ to create three images, which are assigned *three new classes* $y_1, y_2$, and $y_3$, respectively. This results in a larger augmented batch $\tilde{\mathcal{B}}$. Since we generate three new images from each $x$, the size of $\tilde{\mathcal{B}}$ is $8N$. For each original class, we now have 4 classes. For a sample $x \in \tilde{\mathcal{B}}$, let $\tilde{\mathcal{B}}(x) = \tilde{\mathcal{B}} \backslash \{x\}$ and let $P(x) \subset \tilde{\mathcal{B}} \backslash \{x\}$ be a set consisting of the data of the same class as $x$ distinct from $x$. The contrastive representation of a sample $x$ is $z_x = g(h(x,k))/\|g(h(x,k))\|$, where $k$ is the current task. In learning, we minimize the supervised contrastive loss [9] of task $t$.

$$\mathcal{L}_c = \frac{1}{8N} \sum_{x \in \tilde{\mathcal{B}}} \frac{-1}{|P(x)|} \sum_{p \in P(x)} \log \frac{\exp(z_x \cdot z_p/\tau)}{\sum_{x' \in \tilde{\mathcal{B}}(x)} \exp(z_x \cdot z_{x'}/\tau)}, \tag{23}$$

where $\tau$ is a scalar temperature, $\cdot$ is dot product, and $\times$ is multiplication. The loss is reduced by repelling $z$ of different classes and aligning $z$ of the same class more closely. $\mathcal{L}_c$ basically trains a feature extractor with good representations for learning an OOD classifier.

Since the feature extractor is shared across tasks in continual learning, a protection is needed to prevent catastrophic forgetting. HAT and Sup use their respective technique to protect their feature extractor from forgetting. Therefore, the losses $\mathcal{L}$ of Eq. 14 and $\mathcal{L}_{ce}$ of Eq. 22 are replaced by Eq. 23 while the forgetting prevention mechanisms still hold.

### D.3.2 Learning the Classifier.

This is step 2. Given the feature extractor $h$ trained with the loss in Eq. 23, we *freeze* $h$ and only *fine-tune* the linear classifier $f$, which is trained to predict the classes of task $k$ *and* the augmented rotation classes. $f$ maps the feature representation to the label space in $\mathcal{R}^{4|\mathcal{C}^k|}$, where 4 is the number of rotation classes including the original data with $0°$ rotation and $|\mathcal{C}^k|$ is the number of original classes in task $k$. We minimize the cross-entropy loss,

$$\mathcal{L}_{\text{ft}} = -\frac{1}{|\tilde{\mathcal{B}}|} \sum_{(x,y) \in \tilde{\mathcal{B}}} \log \tilde{p}(y|x,k), \tag{24}$$

where ft indicates fine-tune, and

$$\tilde{p}(y|x, tk = \text{softmax}\left(f(h(x,k))\right) \tag{25}$$

where $f(h(x,k)) \in \mathcal{R}^{4|\mathcal{C}^k|}$. The output $f(h(x,k))$ includes the rotation classes. The linear classifier is trained to predict the original *and* the rotation classes. Since individual classifier is trained for each task and the feature extractor is frozen, no protection is necessary.

### D.3.3  Ensemble Class Prediction.

We describe how to predict a label $y \in \mathcal{C}^k$ (TIL) and $y \in \mathcal{C}$ (CIL) ($\mathcal{C}$ is the set of original classes of all tasks). We assume all tasks have been learned and their models are protected by masks.

We discuss the prediction of class label $y$ for a test sample $x$ in the TIL setting first. Note that the network $f \circ h$ in Eq. 25 returns logits for rotation classes (including the original task classes). Note also for each original class label $j_k \in \mathcal{C}^k$ (original classes) of a task $k$, we created three additional rotation classes. For class $j_k$, the classifier $f$ will produce four output values from its four rotation class logits, i.e., $f_{j_k,0}(h(x_0, k))$, $f_{j_k,90}(h(x_{90}, k))$, $f_{j_k,180}(h(x_{180}, k))$, and $f_{j_k,270}(h(x_{270}, k))$, where 0, 90, 180, and 270 represent $0°, 90°, 180°$, and $270°$ rotations respectively and $x_0$ is the original $x$. We compute an ensemble output $f_{j_k}(h(x, k))$ for each class $j_k \in \mathcal{C}^k$ of task $k$,

$$f(h(x, k))_{j_k} = \frac{1}{4} \sum_{\text{deg}} f(h(x_{\text{deg}}, k))_{j_k, \text{deg}}. \tag{26}$$

We use Eq. 9 to make the CIL class prediction, where the final class prediction is made as

$$\hat{y} = \arg\max \bigoplus_i f(h(x, i)). \tag{27}$$

## E  Output Calibration

In this section, we discuss the output calibration technique used in Sec. 4.4 to improve the final prediction accuracy. Even if an OOD detection of each task was perfect (i.e. the model accept and reject IND and OOD samples perfectly), the system could make incorrect class prediction if the magnitudes of outputs across different tasks are different. To ensure that the output values are comparable, we calibrate the outputs by scaling $\alpha_k$ and shifting $\beta_k$ for each task. The optimal parameters $(\alpha_k, \beta_k) \in R \times R$ can be found by solving optimization problem using samples in memory buffer. More precisely, denote the memory buffer $\mathcal{M}$ and calibration parameters $(\alpha, \beta) \in R^T \times R^T$, where $T$ is the number of learned tasks. After training $T$th task, we find optimal calibration parameters by minimizing the cross-entropy loss,

$$\mathcal{L} = -\frac{1}{|\mathcal{M}|} \sum_{(x,y) \in \mathcal{M}} \log p(y|x) \tag{28}$$

where $p(c|x)$ is computed using the softmax,

$$\text{softmax} \bigoplus [\alpha_k f(x)_k + \beta_k] \tag{29}$$

where $\bigoplus$ indicates the concatenation and $f(x)_k$ is the output of task $k$ as Eq. 9. Given the optimal parameters $(\alpha^*, \beta^*)$, we make final prediction as

$$\hat{y} = \arg\max \bigoplus [\alpha_k^* f(x)_k + \beta_k^*] \tag{30}$$

If we use $OOD_k = \sigma(\alpha_k^* f(x)_k + \beta_k^*)$, where $\sigma$ is the sigmoid, and $TP_k = OOD_k / \sum_{k'} OOD_{k'}$, the theoretical results in Sec. 3 hold.

## F  TIL (WP) Results

The TIL (WP) results of all the systems are reported in Tab. 7. HAT and Sup show strong performances compared to the other baselines as they leverage task-specific parameters. However, as shown in Theorem 1, the CIL depends on TP (or OOD). Without an OOD detection mechanism in HAT or Sup, they perform poorly in CIL as shown in the main paper. The contrastive learning in CSI also improves the IND prediction (i.e., WP), and this along with OOD detection results in the strong CIL performance.

## G  Hyper-parameters

Here we report the hyper-parameters that we did not report in the main paper due to space limitations. We mainly report the hyper-parameters of the proposed methods, HAT+CSI, Sup+CSI, and their

Table 7: The TIL results of all the systems. The calibrated versions (+c) of our methods are omitted as calibration does not affect TIL performance. Exemplar-free methods are italicized.

| Method | M-5T | C10-5T | C100-10T | C100-20T | T-5T | T-10T |
|---|---|---|---|---|---|---|
| *OWM* | 99.7±0.03 | 85.0±0.07 | 59.6±0.83 | 65.4±0.48 | 22.4±0.87 | 28.1±0.55 |
| *MUC* | 99.9±0.02 | 95.1±0.10 | 77.3±0.83 | 73.4±9.16 | 55.9±0.26 | 47.2±0.22 |
| *PASS[†]* | 99.5±0.14 | 83.8±0.68 | 72.1±0.70 | 76.8±0.32 | 49.9±0.56 | 46.5±0.39 |
| LwF | 99.9±0.09 | 95.2±0.30 | 86.2±1.00 | 89.0±0.45 | 56.4±0.48 | 55.3±0.35 |
| iCaRL | 99.9±0.08 | 94.9±0.34 | 84.2±1.04 | 85.7±0.68 | 54.5±0.29 | 52.7±0.37 |
| Mnemonics[†*] | 99.9±0.03 | 94.5±0.46 | 82.3±0.30 | 86.2±0.46 | 54.8±0.16 | 52.9±0.66 |
| BiC | 99.9±0.03 | 95.4±0.35 | 84.6±0.48 | 88.7±0.19 | 61.5±0.60 | 62.2±0.45 |
| DER++ | 99.7±0.08 | 92.0±0.54 | 84.0±9.43 | 86.6±9.44 | 57.4±1.31 | 60.0±0.74 |
| *HAT* | 99.9±0.02 | 96.7±0.18 | 84.0±0.23 | 85.0±0.98 | 61.2±0.72 | 63.8±0.41 |
| *HyperNet* | 99.7±0.04 | 94.6±0.37 | 76.8±1.22 | 83.5±0.98 | 23.9±0.60 | 28.0±0.69 |
| *Sup* | 99.6±0.01 | 96.6±0.21 | 87.9±0.27 | 91.6±0.15 | 64.3±0.24 | 68.4±0.22 |
| *HAT+CSI* | 99.9±0.00 | 98.7±0.06 | 92.0±0.37 | 94.3±0.06 | 68.4±0.16 | 72.4±0.21 |
| *Sup+CSI* | 99.0±0.08 | 98.7±0.07 | 93.0±0.13 | 95.3±0.20 | 65.9±0.25 | 74.1±0.28 |

calibrated versions. For all the experiments of the proposed methods, we use the values chosen by the original CSI [3]. We use LARS [10] optimization with learning rate 0.1 for training the feature extractor. We linearly increase the learning rate by 0.1 per epoch for the first 10 epochs. After that, we use cosine scheduler [11] without restart as in [3, 5]. After training the feature extractor, we train the linear classifier for 100 epochs with SGD with learning rate 0.1 and reduce the rate by 0.1 at 60, 75, and 90 epochs. For all the experiments except MNIST, we train the feature extractor for 700 epochs with batch size 128.

For the following hyper-parameters, we use 10% of training data for validation to find a good set of values. For the number of epochs and batch size for MNIST, Sup+CSI trains for 1000 epochs with batch size of 32 while HAT+CSI trains for 700 epochs with batch size of 256. The hard attention regularization penalty $\lambda_i$ in HAT is different by experiments and task $i$. For MNIST, we use $\lambda_1 = 0.25$, and $\lambda_2 = \cdots = \lambda_5 = 0.1$. For C10-5T, we use $\lambda_1 = 1.0$, and $\lambda_2 = \cdots = \lambda_5 = 0.75$. For C100-10T, $\lambda_1 = 1.5$, and $\lambda_2 = \cdots = \lambda_{10} = 1.0$ are used. For C100-20T, $\lambda_1 = 3.5$, and $\lambda_2 = \cdots = \lambda_{20} = 2.5$ are used. For T-5T, $\lambda_i = 0.75$ for all tasks, and lastly, for T-10T, $\lambda_1 = 1.0$, and $\lambda_2 = \cdots = \lambda_{10} = 0.75$ are used. We use larger $\lambda_1$ for the first task than the later tasks as we have found that the larger regularization on the first task results in better accuracy. This is by the definition of regularization in HAT. The earlier task gives lower penalty than later tasks. We manually give larger penalty to the first task. We did not search hyper-parameter $\lambda_t$ for tasks $t \geq 2$. For sparsity in Sup+CSI, we simply choose the least sparsity value of 32 used in the original Sup paper without parameter search.

Calibration methods (HAT+CSI+c and Sup+CSI+c) are based on its memory free versions (i.e. HAT+CSI and Sup+CSI). Therefore, the model training part uses the same hyper-parameters as their calibration free counterparts. For calibration training, we use SGD with learning rate 0.01, 160 training iterations, and batch size of 15 for HAT+CSI+c for all experiments. For Sup+CSI+c, we use the same values for all the experiments except for MNIST. For MNIST, we use learning rate 0.05, batch size of 8, and run 280 iterations.

For the baselines, we use the hyper-parameters reported in the original papers or in their code. If the hyper-parameters are unknown or the code does not reproduce the result (e.g., the baseline did not implement a particular dataset or the code had undergone significant version change), we search for the hyper-parameters as we did for HAT+CSI and Sup+CSI.

## H   Computes and Resources Used in Experiments

This paper provides a guidance on how to solve the CIL problem, backed by theoretical justifications. Based on the guidance, we have proposed some new CIL methods. Two outstanding ones are HAT+CSI and Sup+CSI. These methods achieve state-of-the-art CIL performances, but by no mean,

Table 8: The number of parameters used at inference after learning the final task. The M after each value indicates millions.

| Method | M-5T | C10-5T | C100-10T | C100-20T | T-5T | T-10T |
|--------|------|--------|----------|----------|------|-------|
| OWM | 5.27M | 5.27M | 5.36M | 5.36M | 5.46M | 5.46M |
| MUC | 1.06M | 11.19M | 45.06M | 45.06M | 45.47M | 45.47M |
| PASS | 1.03M | 11.17M | 44.76M | 44.76M | 44.86M | 44.86M |
| LwF | 1.03M | 11.17M | 44.76M | 44.76M | 44.86M | 44.86M |
| iCaRL | 1.03M | 11.17M | 44.76M | 44.76M | 44.86M | 44.86M |
| Mnemonics | 1.03M | 11.17M | 44.76M | 44.76M | 44.86M | 44.86M |
| BiC | 1.03M | 11.17M | 44.76M | 44.76M | 44.86M | 44.86M |
| DER++ | 1.03M | 11.17M | 44.76M | 44.76M | 44.86M | 44.86M |
| HAT | 1.04M | 11.23M | 45.01M | 45.28M | 44.97M | 45.11M |
| HyperNet | 0.48M | 0.47M | 0.47M | 0.47M | 0.48M | 0.48M |
| Sup | 0.05M | 1.43M | 5.75M | 11.45M | 2.95M | 5.80M |
| HAT+CSI | 1.07M | 11.25M | 45.31M | 45.58M | 45.59M | 45.72M |
| HAT+CSI+c | 1.07M | 11.25M | 45.31M | 45.58M | 45.59M | 45.72M |
| Sup+CSI | 0.28M | 1.38M | 5.90M | 11.60M | 3.04M | 6.05M |
| Sup+CSI+c | 0.28M | 1.38M | 5.90M | 11.60M | 3.04M | 6.05M |

they are the only approaches. Many CIL algorithms can be designed following the analysis as it is general to any CL model.

Despite the generality of our work, we report the execution time and required memory for HAT+CSI and Sup+CSI. The report is based on a machine with NVIDIA RTX 3090 on C10-5T experiments. HAT+CSI takes 28.68 hours while Sup+CSI runs for 18.41 hours, which are slower than baselines. Contrastive learning and extensive data augmentation in CSI are the major reason for the slow execution time. However, if other more efficient OOD detection algorithms can replace CSI, the running time can be improved with the new OOD detection methods.

As noted in Sec. 4.2, all the methods use the same backbone architecture with the same width and depth except for OWM and HyperNet for the reasons explained in the main paper. We report the number of parameters of each method required for inference after learning the last task. Sup and Sup+CSI uses a very small number of parameters because Sup finds a sparse subnetwork for each task. Our methods HAT+CSI introduces 7.7K, 17.6K, 68.0K, 47.5K, 191.0K, and 109.0K parameters on M-5T, C10-5T, C100-10T, C100-20T, T-5T, and T-10T, respectively, at each task. Sup+CSI introduces 56.3K, 284.9K, 590.3K, 580.0K, 607.7K, and 605.1K parameters on the same experiments. The calibrated methods HAT+CSI+c and Sup+CSI+c introduce 2 parameters $(\alpha_k, \beta_k)$ per task.

For HAT and HAT+CSI, the reported number of parameters is based on the network at full capacity. The hard attention masks consume 71.10, 86.31, 98.89, 99.71, 92.94, and 98.67% of the total network capacity on average over 5 runs for HAT on M-5T, C10-5T, C100-10T, C100-20T, T-5, and T-10T, respectively. Similarly, 99.39, 99.56, 99.56, 96.74, 94.94, and 99.18% of the total network capacity are used for HAT+CSI on the same datasets on average.

## I  Negative Societal Impacts

The goal of continual learning is to learn a sequence of tasks incrementally. Like many machine learning algorithms, our proposed methods could be affected by bias in the input data as this work does not deal with fairness or bias in the data. A possible solution to mitigate the problem is to check bias in data before training.

## J  Forgetting Rate

We discuss forgetting rate (i.e., backward transfer) [12], which is defined for task $t$ as

$$\mathcal{F}^t = \frac{1}{t-1} \sum_{k=1}^{t-1} \mathcal{A}_k^{\text{init}} - \mathcal{A}_k^t, \tag{31}$$

where $\mathcal{A}_k^{\text{init}}$ is the classification accuracy of task $k$'s data after learning it for the first time and $\mathcal{A}_k^t$ is the accuracy of task $k$'s data after learning task $t$. We report the forgetting rate after learning the last task.

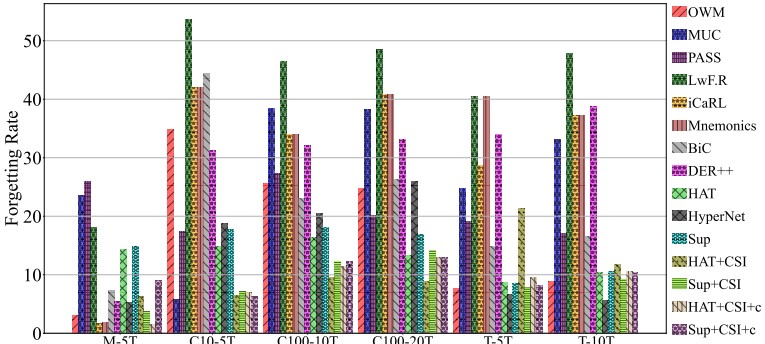

Figure 2: Average forgetting rate (%). The lower the value, the better the method is on forgetting.

Figure 2 shows the forgetting rates of each method. Some methods (e.g., OWM, iCaRL) experience less forgetting than the proposed methods HAT+CSI and Sup+CSI on M-5T. On this dataset, all the systems performed well. For instance, OWM and iCaRL achieve 95.8% and 96.0% accuracy while HAT+CSI and HAT+CSI+c achieve 94.4 and 96.9% accuracy. As we have noted in the main paper, Sup+CSI and Sup+CSI+c achieve only 80.7 and 81.0 on M-5T although they have improved drastically from 70.1% of the base method Sup.

OWM and HyperNet show lower forgetting rates than HAT+CSI+c and Sup+CSI+c on T-5T and T-10T. However, they are not able to adapt to new classes as OWM and HyperNet achieve the classification accuracy of only 10.0% and 7.9%, respectively, on T-5T and 8.6% and 5.3% on T-10T. HAT+CSI+c and Sup+CSI+c achieves 51.7% and 49.2%, respectively, on T-5T and 47.6% and 46.2% on T-10T.

In fact, the performance reduction (i.e., forgetting) in our proposed methods occurs not because the systems forget the previous task knowledge, but because the systems learn more classes and the classification naturally becomes harder. The continual learning mechanisms (HAT and Sup) used in the proposed methods experience little or no forgetting because they find independent subset of parameters for each task, and the learned parameters are not interfered during training. For the forgetting rate results in the TIL setting, refer to our earlier workshop paper [13].