# OpenReview forum: "A Theoretical Study on Solving Continual Learning"
_NeurIPS.cc/2022/Conference — NeurIPS 2022 Accept_

### Official Review · Reviewer_Gwwq · 2022-07-09

**Rating:** 5
**Confidence:** 3
**Soundness:** 2 fair
**Presentation:** 3 good
**Contribution:** 2 fair

**Summary:**

This paper studies class incremental learning(CIL) from a theoretical point of view.
It proposes to decompose CIL into within-task prediction(WIP) and task-id prediction(TP) correlating to out-of-distribution(OOD) detection. It further provides sufficient and necessary conditions for improving CIL performance, and proves that a good performance in WIP, TP, and OOD is necessary for a good performance of CIL. Experiments on a combination of existing CL and OOD methods validate the theoretical results.

**Questions:**

All listed CIL methods are a combination of existing CL and OOD detection methods. It's not clear what are the new designed CIL methods.


**Limitations:**

The authors did not adequately addresses the limitations and potential negative societal impact of the work.

**Strengths And Weaknesses:**

Strengths:
+ The discussion of CIL performance from a theoretical point of view is interesting.

+ The theoretical results provide guidance on selecting a combination of CL and OOD methods for achieving better CIL performance.

+ The authors have conducted extensive experiments on four datasets and nine CL algorithms.

Weaknesses:
- Although the authors claim that they have proposed a new CIL method based on their theoretical analysis. It's not clear what the novelty of the proposed method.

---

> ### Author Response · Authors · 2022-08-02
> **Author response to Reviewer Gwwq**
>
> 1. Although the authors claim that they have proposed a new CIL method based on their theoretical analysis. It's not clear what the novelty of the proposed method.
>
> 	**Reply:** The main contribution of the paper is our theory and its framework, which shows how the CIL problem should be solved. We believe this is very important for the community to know. Although the community has proposed many algorithms to solve the CIL problem, little theory has been proposed. The proposed specific techniques are just some example implementations/methods, but they are also new. Clearly, there can be other methods too, e.g., methods based on some other OOD detection techniques.
>
> 2. All listed CIL methods are a combination of existing CL and OOD detection methods. It's not clear what are the new designed CIL methods.
>
> 	**Reply:** Please also refer to the response to Comment 1 above. It is true that the proposed methods in the paper are combinations of the existing CL and OOD methods, but this type of combination is new as no existing approaches have used this combination. They follow our theoretical result, which is the main contribution of the paper.
>
> 3. The authors did not adequately addresses the limitations and potential negative societal impact of the work.
>
> 	**Reply:** We included some limitations in Conclusion of the paper. Here we elaborate more.
>
>     Limitations: The theoretical analysis is based on the decomposition of CIL probability into within-task in-distribution prediction (WIP) probability and task-id prediction (TP) probability (and OOD probability). It does not cover the data shift that the same data point could be in one class in day one but another class in data two. It also does not handle multi-label classification. Our theory proposes a framework to solve the CIL problem, but it does not propose an optimal algorithm. Following the theoretical framework, there are many options to define WIP and TP (or OOD) in practice, and thus different CIL results.
>
>     In our experiments, we perform class prediction of CIL by simply taking softmax over the output logit values, following the standard CIL prediction, which is a special case of the decomposition as stated in Section 4.2 and Theorem 5 in Appendix. In our future work, we will try to study how to optimize them.
>
>     Societal Impacts: The goal of continual learning is to learn a sequence of tasks incrementally. Like many machine learning algorithms, our proposed methods could be affected by bias in the input data as this work does not deal with fairness or bias in the data. A possible solution to mitigate the problem is to check bias in data before training.
>
>     We have modified Appendix and included the discussion in Appendix H and I.

---

> > ### Comment · Reviewer_Gwwq · 2022-08-06
> > **Keep Score**
> >
> > Thanks for the detailed answer. Most of my concerns are addressed. After reading the other reviews and the answers, I confirm my rating.

---

### Official Review · Reviewer_gU43 · 2022-07-10

**Rating:** 5
**Confidence:** 4
**Soundness:** 2 fair
**Presentation:** 2 fair
**Contribution:** 2 fair

**Summary:**

In this paper, the authors propose that the class-incremental learning problem can be decomposed into two sub-problems: within-task prediction and task-id prediction. They further prove that task-id prediction is correlated to out-of-distribution (OOD) detection. They design some class-incremental learning methods based on their theoretical results and provide experiment results on multiple datasets.

**Questions:**

as the comments are given in weaknesses.

**Limitations:**

not clear in this paper.

**Strengths And Weaknesses:**

Strengths:

- This paper provides in-depth theoretical analyses for class-incremental learning. This is interesting.

- Extensive experiment results on different baselines are provided.

Weaknesses:

- All theoretical analyses are based on the assumption that all classes are independent and identically distributed. However, this might not be true in many real-world applications. For example, there are many classes that correspond to different species of dogs in ImageNet. If these classes occur in different incremental stages, they will significantly influence the task-id prediction. Thus, it is questionable how much these theoretical results can be generalized for real-world applications.

- Some top-performing methods are not cited and compared. For example, [A] archives better performance on CIFAR-100 compared to the proposed method (see Figure 3 in [A]).

- Many details haven’t been provided in the main paper. For example, the authors propose some new CIL techniques in Section 4. However, some important details, such as the number of additional parameters required by the proposed method, are not available. A NeurIPS submission needs to be self-contained, and the authors need to let the audience understander the proposed method without reading the supplementary. It doesn’t seem possible in the current version.

- It would be better to provide a figure to show the overall framework of the proposed method.

- This paper is poorly-organized and difficult to understand. For example, the authors mixed the proposed method with the experiment results in Section 4.

-----

Overall, I think this is an interesting paper with extensive theoretical and experimental results. However, there are some unsolved issues as I mentioned in the “weaknesses” part. Thus, my overall rating is “borderline reject”.

-----

[A] Yan, Shipeng, Jiangwei Xie, and Xuming He. "Der: Dynamically expandable representation for class incremental learning." Proceedings of the IEEE/CVF Conference on Computer Vision and Pattern Recognition. 2021.

---

> ### Author Response · Authors · 2022-08-02
> **Author response to Reviewer gU43**
>
> 1. All theoretical analyses are based on the assumption that all classes are independent and identically distributed. However, this might not be true in many real-world applications. For example, there are many classes that correspond to different species of dogs in ImageNet. If these classes occur in different incremental stages, they will significantly influence the task-id prediction. Thus, it is questionable how much these theoretical results can be generalized for real-world applications.
>
>     **Reply:** If we understand your comment correctly, you meant that each species of dog is a separate class, and different species or classes may appear in the same or different tasks. If that is the case, our theoretical analysis still applies because each dog can only be classified to one species or one class. It is true that as different species of the same animal are usually difficult to distinguish, the prediction will be more difficult, i.e., having lower accuracy (which is the same for humans), but that does not affect our theory. In fact, CIFAR100 and Tiny-ImageNet have many similar classes (e.g., different types of tree), but the experiment shows that our methods are superior to the baselines in these datasets (C100-10T, C100-20T, T-5T, and T-20T). Our work is not applicable to multi-label classification (where a dog may be classified to more than one class) nor hierarchical classification. If our understanding about your comment is incorrect, please let us know and we will be very happy to address it.
>
> 2. Some top-performing methods are not cited and compared. For example, [A] archives better performance on CIFAR-100 compared to the proposed method (see Figure 3 in [A]).
>
> 	**Reply:** DER [A] is a strong replay based method. However, our method is better in most datasets including CIFAR100-10T. The original DER paper reports both the average classification accuracy (ACA) and the average incremental accuracy (AIA). Therefore, we report both results here.
>
> 	ACA
> 	|          | M-5T |  C10-5T | C100-10T | C100-20t | T-5T | T-10T | Avg. |
> 	| ---------| ------| ----- | ----- | ----- | ----- | ----- | ----- |
> 	| DER | 97.2 | 62.1 | 64.5 | 62.5 | 43.6 | 38.3 | 61.4 |
> 	| Sup+CSI+c | 81.0 | 87.3 | 65.2 | 60.5 | 49.2 | 46.2 | 64.9 |
> 	| HAT+CSI+c | 96.9 | 88.0 | 65.2 | 58.0 | 51.7 | 47.6 | 67.9 |
>
> 	AIA
> 	|          | M-5T |  C10-5T | C100-10T | C100-20t | T-5T | T-10T | Avg. |
> 	| ---------| ------| ----- | ----- | ----- | ----- | ----- | ----- |
> 	| DER | 99.0 | 72.1 | 75.4 | 74.1 | 54.2 | 51.2 | 71.0 |
> 	| Sup+CSI+c | 87.5 | 91.7 | 75.4 | 71.2 | 56.2 | 55.9 | 73.0 |
> 	| HAT+CSI+c | 98.3 | 91.9 | 75.9 | 71.0 | 58.6 | 56.5 | 75.4 |
>
>     On average over 6 experiments, the proposed methods Sup+CSI+c and HAT+CSI+c are much stronger than DER (see the last average column). DER achieves 61.4 on ACA while Sup+CSI+c and HAT+CSI+c achieve 64.9 and 67.9, respectively. The gap is similar in AIA. We note that for DER, C10-5T is actually worse than C100-10T and C100-20T, which is unexpected. We did extensive search of optimal learning rate and the number of epochs for DER on C10-5T, and the reported numbers are the best we got. Our method is stronger than DER in most of the experiment settings (C10-5T, C100-10T, T-5T, and T-10T).
>
>     We also note that the main contribution of the paper is the theoretical analysis. The proposed methods are just examples of many possible approaches following the theoretical guidance. They achieve strong performances, but one can easily improve the performance by using a better OOD algorithm according to our theory.
>
> **** Continued on the next response ****

---

> > ### Author Response · Authors · 2022-08-02
> > **Author response to Reviewer gU43**
> >
> > 3. Many details haven’t been provided in the main paper. For example, the authors propose some new CIL techniques in Section 4. However, some important details, such as the number of additional parameters required by the proposed method, are not available. A NeurIPS submission needs to be self-contained, and the authors need to let the audience understander the proposed method without reading the supplementary. It doesn’t seem possible in the current version.
> >
> >     **Reply:** Thanks for the comment. As we have all experienced, due to space limitations, it is really hard to fit all the details in the main paper and we have to put some of them in Appendix in Supplementary Materials. However, since NeurIPS gives an extra content page for the camera-ready version of the accepted papers, we believe we should be able to fit the necessary details in the final version of the paper  if it is accepted.
> >
> >     Regarding the number of parameters, we report the total number of parameters used at inference after learning the final task.
> >
> >     |          | M-5T |  C10-5T | C100-10T | C100-20t | T-5T | T-10T |
> >     | ---------| ------| ----- | ----- | ----- | ----- | ----- |
> >     | DER | 5.2M | 55.9M | 61.6M | 117.6M | 56.4M | 112.7M |
> >     | Sup+CSI | 0.3M | 1.4M | 5.9M | 11.6M | 3.0M | 6.1M |
> >     | Sup+CSI+c | 0.3M | 1.4M | 5.9M | 11.6M | 3.0M | 6.1M |
> >     | HAT+CSI | 1.1M | 11.3M | 44.6M | 44.6M | 44.7M | 44.7M |
> >     | HAT+CSI+c | 1.1M | 11.3M | 44.6M | 44.6M | 44.7M | 44.7M |
> >
> >     The proposed methods Sup+CSI and HAT+CSI use much smaller number of parameters at inference than DER.
> >
> >     At each task, Sup+CSI introduces 56.3K, 284.9K, 590.3K, 580.0K, 607.7K, and 605.1K parameters on M-5T, C10-5T, C100-10T, C100-20T, T-5T, T-10T, respectively, while HAT+CSI introduces 7.7K, 17.6K, 68.0K, 47.5K, 191.0K, and 109.0K parameters. The calibrated methods Sup+CSI+c and HAT+CSI+c introduces 2 parameters per task. Please refer to Appendix G in the revised version for the number of parameters of all the baselines.
> >
> > 4. It would be better to provide a figure to show the overall framework of the proposed method.
> >
> >     **Reply:** Thanks for the suggestion. We have included a figure explaining the overall framework in the updated Appendix. Please refer to Appendix D.
> >
> > 5. This paper is poorly-organized and difficult to understand. For example, the authors mixed the proposed method with the experiment results in Section 4.
> >
> >     **Reply:** Our apology if our presentation was confusing. Since the main contribution of the paper is the theoretical analysis, the proposed methods are just some possible examples, although they are also new (clearly, there can be other possible methods, e.g., based on some other OOD techniques). That is why we put them in the experiment section. We will put the methods in a separate section before the experiment section. We will do the revision in the next version as this change will be drastic and can be disruptive to your reading and that of other reviewers as you and other reviewers are more familiar with the original submitted version.

---

> > > ### Comment · Reviewer_gU43 · 2022-08-07
> > > **after rebuttal**
> > >
> > > Thanks for the careful replies. I am ok with most of them, and thus promote my rating by 1.

---

> > > > ### Author Response · Authors · 2022-08-07
> > > > **Author response to Reviewer gU43**
> > > >
> > > > Thank you so much. We really appreciate it.

---

### Official Review · Reviewer_2nNb · 2022-07-12

**Rating:** 5
**Confidence:** 4
**Soundness:** 3 good
**Presentation:** 3 good
**Contribution:** 3 good

**Summary:**

The paper studies Class Incremental Learning (CIL) and shows that the CIL problem can be decomposed into two subproblems: within-task in-distribution prediction (WIP) and task-id prediction (TP). As a result, the CIL performance is bounded by WIP and TP performances. Moreover, the authors show that the TP subproblem itself can be bounded by the OOD detection problem, and thus by extension, CIL can be bounded by WIP and OOD.

To test their theory, the authors add better OOD detection to several CL algorithms and report performance improvements.

**Questions:**

See above.

**Limitations:**

Adequate discussion.

**Strengths And Weaknesses:**

**Strengths**:


1. The paper is very well-organized and well-written. It is easy to follow the arguments.


2. The analysis is mostly intuitive, and I appreciate the effort to understand CIL more theoretically.




**Weaknesses**:


1. I'm not entirely convinced about the bounds in the paper, and I think the theory is partially incomplete. Let's assume we have an oracle that, given a data point, can tell us the task id with zero loss. Also, let's assume if we train on each task individually, we have zero loss. Then by Theorem 1, it means that we can have zero CIL loss. However, this implies that we should have zero loss for the equivalent TIL problem (since we have task ids), which is not necessarily true. I think (but not sure) the theory holds if we assume we don't have shared parameters for tasks (e.g., parameter isolation methods) or tasks are all orthogonal. Otherwise, even if we know about task ids (e.g., TIL), we still can show the error cannot be bounded. As an example, let's say we have two tasks and we always have access to task ids, but we use the same parameters for both tasks. Then if we train on task 2 for a very long time (without using any CL algorithm), since our loss function only depends on the data of task 2, and since we have shared parameters, the parameters will change, and if we measure the loss on task 1, it can grow (similar to all other TIL works), unless we have additional assumptions. I am willing to discuss this further with the authors during the discussion period.


2. Discrete Task Assumption: Assumptions 1 and 2 assume that each data point belongs to one task exactly and tasks are discrete. However, in many real-world setups, the shift may be continuous.



3. I have read the argument on Sec.4.2 (L274), but I still think the forgetting rate is an important metric that measures the backward transfer in many algorithms. I encourage the authors to compare the average forgetting for a few benchmarks.



Overall, while I think this work is a nice first step, I think the theory is limited in its correct form, and more importantly, I think we need additional assumptions. I am willing to raise my score if my concerns are addressed.



--------
Update:
After discussion with the authors, I believe they have addressed my concerns, and I raise the score.

---

> ### Author Response · Authors · 2022-08-02
> **Author response to Reviewer 2nNb**
>
> 1. I'm not entirely convinced about the bounds in the paper, and I think the theory is partially incomplete. Let's assume we have an oracle that, given a data point, can tell us the task id with zero loss. Also, let's assume if we train on each task individually, we have zero loss. Then by Theorem 1, it means that we can have zero CIL loss. However, this implies that we should have zero loss for the equivalent TIL problem (since we have task ids), which is not necessarily true. I think (but not sure) the theory holds if we assume we don't have shared parameters for tasks (e.g., parameter isolation methods) or tasks are all orthogonal. Otherwise, even if we know about task ids (e.g., TIL), we still can show the error cannot be bounded. As an example, let's say we have two tasks and we always have access to task ids, but we use the same parameters for both tasks. Then if we train on task 2 for a very long time (without using any CL algorithm), since our loss function only depends on the data of task 2, and since we have shared parameters, the parameters will change, and if we measure the loss on task 1, it can grow (similar to all other TIL works), unless we have additional assumptions. I am willing to discuss this further with the authors during the discussion period.
>
> 	**Reply:** Thank you for your effort to carefully check our theorems. We believe there are two misunderstandings here. The first is that CIL = WIP * TP in Eq.2 means that when we have WIP and TP (defined either explicitly or implicitly by implementation), we can find a corresponding CIL defined by WIP * TP. Similarly, when we have a CIL, we can find the corresponding underlying WIP and TP defined by their probabilistic definition. For the oracle example, if we can predict TP with 100% precision (or zero loss since we have task ids) and also predict WIP with 100% precision (or zero loss), we have already had a CIL by simply taking WIP * TP, whose corresponding CIL achieves zero loss.
>
> 	The second is that as stated in the paper (lines 77-78, footnote 3, lines 135-137, line 166-167, and lines 203-204), our theory is about after-training rather than intermediate-training or any training algorithm. Here after-training means any time when the CIL model can be used for predicting of test cases. So, after finishing training, we measure the performance of WIP and TP and show that CIL is bounded by WIP and TP. In our theory, we don't care about the training process but only the final model. Hence, our theory doesn't care if the algorithm isolates parameters or shares parameters across tasks. It only claims that any trained CIL model must satisfy the conditions (Theorem 4) to perform good CIL. In your example, since task 2 is trained more and task 1's WIP will be poor, our theory tells that the final CIL on task 1 will be very poor.
>
> 	We understand that this may be involved. Please let us know if you need a further discussion.
>
> 2. Discrete Task Assumption: Assumptions 1 and 2 assume that each data point belongs to one task exactly and tasks are discrete. However, in many real-world setups, the shift may be continuous.
>
> 	**Reply:** Our theory is based on the existing definition of class-incremental learning (CIL). It assumes that each data point belongs to a single class and a single task as there are no overlapping classes among tasks.
> 	By construction, tasks are disjoint. As there are many types of data shift, we are not sure which specific types of data shift you refer to. If you meant that a learned class in a task may appear again later, then our theory does cover that. Suppose dataset 1 has classes {dog, cat, tiger} and dataset 2 has classes {dog, computer, car}. We can define task 1 as {dog, cat, tiger} and task 2 as {computer, car}. The shared class *dog* in dataset 2 can be regarded as additional training data of dog has appeared after task 1. Note also (see our response to your first question above) that our theory is only applied to the CIL model after-training. It does not care about the training process, the training algorithm, or when any part of the training data of each class appears.
>
> 	This paper does not deal with the type of data shift that a data point belongs to one class in day one and a different class in day two. For this type of data shifting scenario, forgetting will not be so much of an issue because the past knowledge may no longer be valid. We also do not deal with multi-label classification, in which a data point may belong to more than one class. We plan to study different types of data shift and their impact on continual learning in our future work.
>
> **** Continued on the next response ****

---

> > ### Author Response · Authors · 2022-08-02
> > **Author response to Reviewer 2nNb**
> >
> > 3. I have read the argument on Sec.4.2 (L274), but I still think the forgetting rate is an important metric that measures the backward transfer in many algorithms. I encourage the authors to compare the average forgetting for a few benchmarks.
> >
> > 	**Reply:** Following your suggestion, we have included a discussion of backward transfer [1] in Appendix J.
> > 	Here, we report the average forgetting rates of the best performing baselines: Mnemonics, BiC, and DER++. The three methods experience forgetting of 42.1, 44.4, and 31.2, respectively on C10-5T and 33.9, 23.0, and 32.2 on C100-10T. Our proposed memory free methods, HAT+CSI and Sup+CSI, achieve 6.4 and 7.1 on C10-5T and 9.5 and 12.3 on C100-10T. Note that the average classification accuracy of the three baselines are 65.1, 61.4, and 66.0, respectively, on C10-5T while the proposed methods achieve 87.7 and 86.0, respectively, on the same experiment. On C100-10T, the three methods achieve 51.0, 52.9, and 53.7, respectively. Our methods achieve 63.3 and 65.1, respectively. The forgetting rate and the classification accuracy show that our methods are good at preserving the previous knowledge while adapting to new classes.
> >
> > 	Please refer to the updated Appendix J for the results of all the baselines on other datasets.
> >
> > [1] David Lopez-Paz and Marc’Aurelio Ranzato. Gradient Episodic Memory for Continual Learning. In NeurIPS, 2017.

---

> > > ### Comment · Reviewer_2nNb · 2022-08-05
> > > **Response to authors**
> > >
> > > I would like to thank the authors for addressing my questions.
> > >
> > > 1. Regarding the theory, I apologize for the misunderstanding and I agree with your reasoning. Hence, I will raise my score. But, I think solving WIP is still a very difficult problem to overcome. If we have an oracle for TP with zero loss, the CIL problem will be reduced to TIL (task-incremental problem) and we know that we cannot still solve the TP problem due to many factors such as catastrophic forgetting. While this paper does a nice job formalizing this relationship, still, we do not know how to solve the problem.
> > >
> > > 2. Thanks for your explanation on the discrete task assumption. What I meant was that we may not know the task identifiers when we deal with a stream of data. However, I agree with your argument.
> > >
> > > 3. Thanks for reporting the forgetting metrics. As I explained, this is an important metric and it will be helpful for follow-up works.
> > >
> > >
> > > Overall, I am satisfied with the authors' response and I would like to thank them again.

---

> > > > ### Author Response · Authors · 2022-08-06
> > > > **Author response to Reviewer 2nNb**
> > > >
> > > > Thank you very much. We really appreciate it.

---

### Official Review · Reviewer_SV1o · 2022-07-13

**Rating:** 6
**Confidence:** 3
**Soundness:** 3 good
**Presentation:** 3 good
**Contribution:** 3 good

**Summary:**

The paper investigates the Class incremental learning (CIL) problem by expressing the problem as a Within-task in-distribution (IND) class learning problem (WIP) and task-id prediction (TP) problem. The author theoretically induces that solving the two problems: WIP and TP, is a necessarily sufficient condition for solving CIL, and also shows that TP is closely related to the out-of-distribution detection (OOD) problem. Based on the investigation, the author proposes a new CIL technique using recent OOD algorithms (solving WIP and OOD simultaneously) and achieved superior CIL performance, supporting the author's claim that OOD is a critical factor for CIL.

**Questions:**

See the Strongpoints/weakness page.

**Limitations:**

See the Strongpoints/weakness page.

**Strengths And Weaknesses:**

StrongPoints

(1) The theoretical investigation of CIL, starting from WIP and TP to OOD, seems reasonable and worth sharing with the community.

(2) The proposed simple CIL solution using OOD supports the claim of the author, that OOD which can also solve WIP, can sufficiently solve CIL problem. Moreover, comparision to many existing CIL problems increases the credibility of the author's suggestion.

WeakPoints

First, note that the reviewer is positive about this work, and this section is much like questions rather than criticism.

(1) The paper focuses on the traditional CIL problem set-up that the incoming data label is completely disjoint. However, a recent line of studies also introduces a more realistic, close-to real-world setting, where the incoming data label is overlapped, which called Blurry task setting [1][2]. When following the author's theoretical induction, the reviewer could not find specific problems restricting the induction to be applied in the Blurry task setting. Hence, applying the idea to the Blurry set-up will strengthen the author's claim.

[1] Rainbow memory: Continual learning with a memory of diverse samples. J Bang, H Kim, YJ Yoo, JW Ha, J Choi - Proceedings of the IEEE/CVF Conference on Computer Vision and Pattern Recognition, 2021
[2] Online Continual Learning on a Contaminated Data Stream with Blurry Task Boundaries. J Bang, H Koh, S Park, H Song, JW Ha, J Choi - Proceedings of the IEEE/CVF Conference on Computer Vision and Pattern Recognition, 2022

(2) For the testing datasets, recent CIL problems also use ImageNet to show their algorithm's performance larger than CIFAR 10/100 or tiny-ImageNet. The reviewer does not think that the tendency shown in the experiments will not change drastically, but checking it in ImageNet would much strengthen the credibility of the Experiments.

---

> ### Author Response · Authors · 2022-08-02
> **Author response to Reviewer SV1o**
>
> 1. The paper focuses on the traditional CIL problem set-up that the incoming data label is completely disjoint. However, a recent line of studies also introduces a more realistic, close-to real-world setting, where the incoming data label is overlapped, which called Blurry task setting [1][2]. When following the author's theoretical induction, the reviewer could not find specific problems restricting the induction to be applied in the Blurry task setting. Hence, applying the idea to the Blurry set-up will strengthen the author's claim.
>
> 	**Reply:** We believe there is a misunderstanding. Our theory is applicable to the blurry task setting. Footnote 3 actually tries to state that but it was not sufficiently detailed.
>
> 	In the blurry task setting, different sets of data (e.g., tasks) have shared classes. Suppose dataset 1 has classes {dog, cat, tiger} and dataset 2 has classes {dog, computer, car}. We can define task 1 as {dog, cat, tiger} and task 2 as {computer, car}. The shared class *dog* in dataset 2 can be regarded as additional training data of dog appeared after task 1. We can do this because, as we stated in the paper, our theory is applied only to the CIL model after training. It does not care how the training was done or when any part of the training data of each class appears, i.e., it tells what a trained CIL model should satisfy in order to produce good performances.
>
> 	As a further explanation, we can look at the extreme case where each task consists of only a single class and any portion of the training data of each class can appear any time (i.e., any class can be revisited). Our theory clearly holds. This is because in this case, after training, the within-task in-distribution prediction (WIP) becomes trivial (i.e., the probability of 1) as there is only a single class to predict within each task. The task-id prediction (TP) becomes class-id prediction as the number of tasks to predict is the same as the number of classes.
>
> 	So our theory is applicable as long as each data point belongs to a single class and a single class only as tasks are actually artificial in CIL.
>
> 	Hope this explanation is clear. Please let us know if you still have doubts.
>
>
> 2. For the testing datasets, recent CIL problems also use ImageNet to show their algorithm's performance larger than CIFAR 10/100 or tiny-ImageNet. The reviewer does not think that the tendency shown in the experiments will not change drastically, but checking it in ImageNet would much strengthen the credibility of the Experiments.
>
> 	***Reply:*** That is correct. Since our analysis is not based on the dataset and is general to any CIL problem, the tendency shown in the experiments does not change by dataset.
>
> 	We did not conduct experiments on larger dataset such as ImageNet because all the baselines already performed very poorly on TinyImageNet. However, taking your comment, we have conducted additional experiment on a subset of ImageNet instead of the full-ImageNet due to the limited time in preparing the rebuttal. We randomly selected 400 classes from the original 1000 classes, and chose 50% of training samples per class. We split 400 classes into 10 tasks with 40 classes per task.
> 	We ran the best performing baselines DER++ and HAT and applied the post-processing OOD detection method ODIN. The original DER++ achieves 20.37 classification accuracy with AUC 67.15. Applying ODIN, the CIL improves to 21.92 as AUC improves 68.26. HAT without ODIN achieves 27.24 classification accuracy and 69.41 AUC. After ODIN, HAT achieves 27.49 accuracy and 69.51 AUC. The improvement is not large because as we noted in the paper, the OOD detection method ODIN is quite weak, but the tendency remains consistent as our theorem says. Using stronger OOD detection methods would improve the performance further. We could not run our stronger methods HAT+CSI or Sup+CSI because the OOD detection method CSI is based on class/data augmentation and contrastive learning, which takes time to run. In practice, a more efficient OOD detection method can be chosen. We also note that the main contribution of the paper is the theory. The proposed methods (which are also new) are just some example techniques that follow the theory. Clearly, other methods can be proposed based on our theory, e.g., using stronger and/or more efficient ODD methods.

---

### Meta-Review · Area_Chair_CtTi · 2022-08-26

**Recommendation:** Accept
**Confidence:** Certain

**Metareview:**

This paper studies the class-incremental setting of continual learning. The authors provide a theoretical justification for decomposing the problem in two: a) task-id prediction, and b) within-task prediction. Based on this insight, the authors propose two new continual-learning methods.

The initial reviews were somewhat mixed and the reviewers raised several questions, especially on some of the theoretical arguments proposed in the paper. The authors provided detailed replies that were convincing to the reviewers and the reviewers now agree that this is a good paper that should be accepted. I would add that this contribution might prove significant as most existing CL algorithms are empirical in nature rather than grounded in theory. This is likely even more so for the more challenging class-incremental learning setting.

I am happy to recommend acceptance of this paper, congratulations! I also strongly encourage the authors to take into account the comments of the reviewers when preparing the camera-ready version of this work. In particular:
- Several expert reviewers misunderstood key aspects of the paper upon first reviewing it. I do not have precise suggestions for improving the paper, but it would be useful to explicitly discuss some points raised by the reviewers in the paper or in the appendix. There are many different flavors of continual learning even within class-incremental and so clarifying the exact scope of the theoretical development might also help.
- In your replies to the reviewers, you mention not having enough time to study your methods on imagenet. Regardless of the performance of tiny-imagenet, my sense is that using the full imagenet would improve the (perceived) significance of the results.


**Award:**

No

---

### Decision · Program_Chairs · 2022-09-14

Accept